# JADE: Joint Alignment and Deep Embedding for Multi-Slice Spatial Transcriptomics

**Yuanchuan Guo**
Department of Statistics
Harvard University
yguo1@g.harvard.edu

**Jun S. Liu**[*]
Department of Statistics
Tsinghua University
junsliu@tsinghua.edu.cn

**Huimin Cheng**[*]
Department of Biostatistics
Boston University
huimin23@bu.edu

**Ying Ma**[*]
Department of Biostatistics
Center for Computational Molecular Biology
Brown University
ying_ma@brown.edu

## Abstract

As spatially resolved transcriptomics (SRT) datasets increasingly span multiple adjacent or replicated slices, effective joint analysis across slices is needed to reconstruct tissue structures and identify consistent spatial gene expression patterns. This requires resolving spatial correspondences between slices while capturing shared transcriptomic features, two tasks that are typically addressed in isolation. Multi-slice analysis remains challenging due to physical distortions, technical variability, and batch effects. To address these challenges, we introduce Joint Alignment and Deep Embedding for multi-slice SRT (JADE), a unified computational framework that simultaneously learns spatial location-wise alignments and shared low-dimensional embeddings across tissue slices. Unlike existing methods, JADE adopts a roundtrip framework in which each iteration alternates between alignment and embedding refinement. To infer alignment, we employ attention mechanisms that dynamically assess and weight the importance of different embedding dimensions, allowing the model to focus on the most alignment-relevant features while suppressing noise. To the best of our knowledge, JADE is the first method that jointly optimizes alignment and representation learning in a shared latent space, enabling robust multi-slice integration. We demonstrate that JADE outperforms existing alignment and embedding methods across multiple evaluation metrics in the 10x Visium human dorsolateral prefrontal cortex (DLPFC) and Stereo-seq axolotl brain datasets. By bridging spatial alignment and feature integration, JADE provides a scalable and accurate solution for cross-slice analysis of SRT data.

## 1 Introduction

Spatially resolved transcriptomics (SRT) technologies provide high-throughput measurements of gene expression within tissue sections while preserving spatial context [6, 44, 48, 61, 29]. By enabling the joint study of molecular states and tissue architecture, SRT has transformed our understanding of developmental processes [9, 72], disease microenvironments [47, 83], and spatial cellular organization across diverse biological systems, from cancer [56, 5, 60, 27, 47, 1, 16] to neuroscience [43, 45, 53, 77]. As the resolution and scale of SRT continue to improve, there is growing interest in multi-slice SRT datasets, where multiple adjacent or replicated tissue sections are profiled to reconstruct 3D structures, map spatial trajectories, or assess reproducibility across individuals and conditions [19, 57, 21, 42, 14, 31, 41, 35]. However, analyzing multi-slice SRT data introduces a set of unique

---

[*]Co-corresponding authors

39th Conference on Neural Information Processing Systems (NeurIPS 2025).

challenges. Each tissue slice may undergo non-linear spatial deformation during sectioning and mounting, while also exhibiting substantial variation in transcript capture efficiency and local tissue composition [79, 37]. These factors confound the alignment of corresponding regions across slices and obscure shared biological signals. Effective multi-slice integration must therefore address two tightly coupled tasks: spatial alignment and representation learning [39, 67, 36]. Alignment resolves spatial location -level correspondences across slices, while representation learning compresses high-dimensional gene expression data into a shared latent space that supports robust downstream analysis. Despite their interconnected nature, existing methods typically address these tasks in isolation, limiting their ability to perform coherent, biologically meaningful integration.

Previous approaches fall into two broad categories. Alignment-based methods [37, 75, 12, 28, 20], such as PASTE [79], estimate spatial location - level mappings across adjacent slices by jointly considering spatial coordinates and gene expression similarity, enabling reconstruction of 3D tissue volumes. However, these methods operate directly on raw expression profiles, which are often sparse, noisy, and affected by batch effects. These batch effects refer to systematic technical variations introduced during sample processing or sequencing, which can obscure true biological signals. As a result, they do not provide low-dimensional representations that are essential to downstream tasks such as spatial domain detection or trajectory inference. Conversely, representation learning–based methods [22, 34, 82, 80, 52, 23, 7, 74, 78], such as GraphST [38] and STAGATE [15], leverage graph neural networks to extract informative low-dimensional embeddings. While effective for extracting latent embeddings and identifying spatial domains, these methods typically process slices independently or jointly without explicitly resolving anatomical correspondences. As a result, homologous tissue regions may be represented inconsistently across slices, undermining interpretability and cross-slice comparison. To address the need for joint analysis across samples, integration methods originally developed for single-cell RNA sequencing data [73, 18], including Harmony [33], Seurat [57], and scVI [40], do not account for spatial context and assume shared coordinate systems across samples, assumptions that rarely hold in spatial data. Recently, approaches such as STAligner [81] and PRE-CAST [36] have extended integration techniques to spatial transcriptomics, but they either require additional input (e.g., batch id or histology image) or do not explicitly model spatial location-level alignment. These methods focus on harmonizing latent features across slices but cannot account for physical tissue distortions that are critical for spatial reconstruction.

Together, these limitations highlight the need for a unified framework that can resolve spatial correspondences and learn biologically meaningful representations across multiple slices, allowing alignment to guide representation learning, and vice versa. To address this need, we introduce JADE (Joint Alignment and Deep Embedding), a computational framework that integrates multi-slice SRT data by simultaneously learning (1) a probabilistic alignment between spatial locations and (2) a shared low-dimensional embedding space. JADE performs alignment in the latent space via attention-based optimal transport and enforces spatial and transcriptomic consistency through graph-based contrastive learning. This coupling ensures that learned embeddings are mutually aligned and biologically coherent, while correspondences between spatial locations across tissue slices respect both spatial geometry and gene expression structure.

To the best of our knowledge, JADE is the first method to jointly optimize spatial location-level alignment and representation learning within a unified, spatially informed model. We benchmark JADE on multi-slice SRT datasets from the human dorsolateral prefrontal cortex (DLPFC) [45] and the regenerating axolotl brain [68], and show that it consistently outperforms state-of-the-art alignment and embedding methods in spatial clustering accuracy, alignment fidelity, and biological interpretability. JADE offers a scalable and robust solution for integrative spatial analysis, particularly in settings that require the joint resolution of anatomical correspondence and functional representation across complex tissue landscapes.

## 2 Methods

**Problem Definition.** Before we present our method JADE, we first formally introduce the problem setup. Consider a pair of SRT slices $(S_1, X_1)$ and $(S_2, X_2)$, where $S_1 \in \mathbb{R}^{n_1 \times 2}$ and $S_2 \in \mathbb{R}^{n_2 \times 2}$ represent the spatial coordinates of $n_1$ and $n_2$ spatial locations in the two tissue slices, and $X_1 \in \mathbb{R}^{n_1 \times p}$ and $X_2 \in \mathbb{R}^{n_2 \times p}$ correspond to their respective gene expression matrices, where $p$ represents the same set of genes measured across tissue slices. Given both gene expression and spatial coordinates for the two slices, our objective is to jointly learn: (1) an alignment matrix $\Pi \in \mathbb{R}^{n_1 \times n_2}$ between

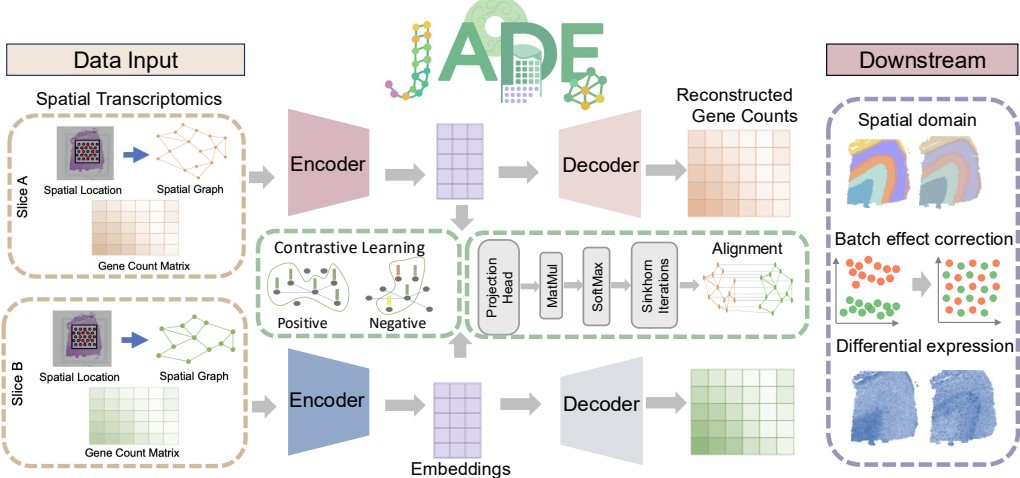

**Figure 1:** Workflow of JADE. Given two slices of SRT data (left box), JADE learns a probabilistic alignment and low-dimensional embeddings simultaneously. One unique feature of JADE is that embedding and alignment are updated in a roundtrip way in which each iteration alternates between alignment and embedding refinement. Downstream applications (right box) include spatial domain detection, batch effect correction, and differential expression analysis.

these spatial locations, where $\Pi_{ij}$ encodes the correspondence between spatial location $i$ in slice 1 and spatial location $j$ in slice 2, and (2) the low-dimensional embedding representations $H_1$ and $H_2$.

**Overview of JADE.** Figure 1 illustrates the workflow of JADE. The pipeline begins with the input of SRT data from two slices (left panel). These inputs are passed through encoders to extract low-dimensional embeddings. Simultaneously, JADE infers cross-slice correspondences via a graph attention module to obtain embedding-space alignment. This module consists of a projection head, matrix multiplication, softmax, and iterative sinkhorn operations. To enhance generalizability, we employ a contrastive learning module that maximizes agreement between spatially neighboring locations while distinguishing dissimilar ones. A key innovation of JADE is its roundtrip learning scheme, in which embeddings and alignments are refined alternately within each training iteration. The outputs of JADE include a probabilistic alignment matrix between spatial locations across two tissue slices and embedding representation of each spot. These outputs support downstream analyses such as 3D tissue reconstruction (via the alignment matrix), spatial domain detection (via the learned embeddings), batch effect correction, and differential expression analysis. Detailed descriptions of each module are provided in the following sections and the pseudo code is shown in Appendix A. The implementation of JADE, along with data preprocessing scripts and pretrained models, is publicly available at https://github.com/YMa-lab/JADE.

### 2.1 Graph Autoencoder

Let $A_1 \in \mathbb{R}^{n_1 \times n_1}$, and $A_2 \in \mathbb{R}^{n_2 \times n_2}$ denote the two adjacency matrices of the spatial graph in two slices obtained using the method described in Appendix B. Given two SRT slices $(A_1, X_1)$ and $(A_2, X_2)$, we first encode them independently using graph convolutional networks (GCNs) [32] to obtain initial embeddings. Each encoder aggregates information from neighboring spatial locations to learn low-dimensional embeddings using a one-layer GCN: $H_1 = \text{Relu}(\tilde{A}_1 X_1 W_{1e} + b_{1e})$, $H_2 = \text{Relu}(\tilde{A}_2 X_2 W_{2e} + b_{2e})$, where $\tilde{A}_i = D_i^{-\frac{1}{2}} A_i D_i^{-\frac{1}{2}}$ is the degree normalized adjacency matrix, ensuring proper feature scaling across neighboring locations. Without normalization, a node with more neighbors would accumulate disproportionately large feature values, leading to biased representations. $W_{ie}$ and $b_{ie}$ are trainable weight and bias parameters of the encoder for slice $i \in \{1, 2\}$. ReLU is applied as a nonlinear activation function to enhance feature expressiveness. $H_1 \in \mathbb{R}^{n_1 \times d}$ and $H_2 \in \mathbb{R}^{n_2 \times d}$ are the latent embeddings of two slices, where $d$ is a hyperparameter that defines the latent dimension.

To ensure that the learned embeddings retain key biological information, we decode them back into the original feature space using a GCN-based decoder: $\hat{X}_1 = \text{Relu}(\tilde{A}_1 H_1 W_{1d} + b_{1d})$, $\hat{X}_2 = \text{Relu}(\tilde{A}_2 H_2 W_{1d} + b_{1d})$, where $W_{id}$ and $b_{id}$ represent trainable parameters of the decoder for slice

$i \in \{1, 2\}$. The reconstruction loss ensures that the decoded outputs $\hat{X}_1, \hat{X}_2$ accurately recover the original gene expression profiles $\mathcal{L}_{\text{recon}} = \frac{1}{n_1}||X_1 - \hat{X}_1||_F^2 + \frac{1}{n_2}||X_2 - \hat{X}_2||_F^2$, where $|| \cdot ||_F$ denotes the Frobenius norm, measuring the difference between the reconstructed and original feature matrices.

## 2.2 Graph Attention for Embedding-Space Alignment

Given embeddings $H_1, H_2$, we obtain the alignment matrix $\Pi$ through the following steps.

**Step 1: Compute cross-attention weights.** We first compute an attention-based similarity matrix $C \in [0, 1]^{n_1 \times n_2}$ to measure correspondences between embeddings of two slices:

$$C = \text{Softmax}\left(\frac{-H_1^T M M^T H_2}{\sqrt{d}}\right). \tag{1}$$

Here $M \in \mathbb{R}^{d \times d}$ is a learnable linear projection implementing an attention mechanism [64], enabling the model to focus on key cross-slice features.

**Step 2: Obtain the alignment matrix from cross-attention weights.** The cross-attention weights $C$ encode pairwise embedding similarities, but it is only row-stochastic: each row sums to 1, yet the column totals are unrestricted. This imbalance can lead to biased mappings, where some target locations accumulate disproportionately high mass, while others receive very little.

To achieve a balanced alignment, we reinterpret the alignment problem within an *optimal transport* (OT) [62] framework, enforcing both rows and columns to satisfy uniform marginal constraints. In this framework, the alignment matrix satisfy $\Pi \in [0, 1]^{n_1 \times n_2}$, $\sum_{i,j} \Pi_{ij} = 1$. Each element $\Pi_{ij}$ denotes the amount of probability mass transported from source point $i$ to target point $j$, with the constraint that the entire matrix satisfies specific marginal distributions. Specifically, we apply the Sinkhorn-Knopp algorithm [55, 13] to transform the raw similarity scores in $C$ into a doubly-stochastic alignment matrix $\Pi$. This iterative procedure alternately normalizes rows and columns, enforcing each row sum to $1/n_1$ and each column sum to $n_2$, ensuring that every spatial location contributes and receives an equal amount of alignment mass. Thus, the alignment becomes fair and balanced, avoiding situations where a few locations dominate the mappings. Additionally, we introduce a marginal regularization term into our objective function: $\mathcal{L}_{\text{marginal}} = \text{KL}\left(\frac{\mathbf{1}_n}{n_1} \cdot \Pi \, || \, \frac{\mathbf{1}_{n_2}}{n_2}\right)$. This penalty is defined as the KL divergence between the column sums of $\Pi$ and the desired uniform marginal distribution. In doing so, we explicitly discourage deviations from uniformity, further reinforcing balanced alignments.

**Step 3: Spatial and embedding aware alignment losses.** To refine the alignment and maintain spatial structure, we define two primary losses motivated by the fused Gromov-Wasserstein distance [46, 65]: (1) Spatial structure preservation loss (Mis-Maintain Loss) ensures that spatial relationships in both slices remain consistent after alignment $\mathcal{L}_{\text{maintain}} = \frac{1}{n_1}||D_1 - n_2^2 \Pi D_2 \Pi^T||_F + \frac{1}{n_2}||D_2 - n_1^2 \Pi^T D_1 \Pi||_F$, where $D_1$ and $D_2$ are pairwise spatial distance matrices. Each entry $d_{1ij}$ (for slice 1) and $d_{2ij}$ (for slice 2) represents the squared Euclidean distance between spatial locations $i$ and $j$. The scaling factors $n_2^2$ and $n_1^2$ ensure that the transported distance matrices $\Pi D_2 \Pi^T$ and $\Pi^T D_1 \Pi$ remain on the same scale as the original $D_1$ and $D_2$, compensating for the normalization inherent in $\Pi$. Minimizing $\mathcal{L}_{\text{maintain}}$ encourages spatial locations that were close to each other before alignment to remain close after alignment, while spatial locations that were far apart should continue to be far apart. (2) Embedding alignment loss (Mis-Alignment Loss) ensures that the aligned embeddings are close to each other, maintaining meaningful biological correspondence $\mathcal{L}_{\text{align}} = \frac{1}{n_1}||H_1 - n_2 \Pi H_2||_F + \frac{1}{n_2}||H_2 - n_1 \Pi^T H_1||_F$. Similarly, the scaling factors $n_2$ and $n_1$ ensure that the transported embedding matrices $n_2 \Pi H_2$ and $n_1 \Pi^T H_1$ remain on the same scale as the original $H_1$ and $H_2$. The first term in the above equation minimizes the discrepancy between slice 1 embeddings and their aligned counterparts from slice 2. The second term enforces the same alignment constraint in the reverse direction.

## 2.3 Self-supervised Graph Contrastive Learning

We refine our embeddings via self-supervised contrastive learning similar to GraphST [38], using the Graph Infomax objective [66] to maximize mutual information between each spot and its local neighborhood. This drives adjacent, biologically similar spots closer in latent space while pushing structurally distinct or distant spots apart.

For each spatial location $i$, we define its local spatial proxy $r_i$ as: $r_i = \frac{1}{|N(i)|} \sum_{j \in N(i)} h_j$, where $N(i)$ denotes the set of immediate neighbors of spatial location $i$, $h_j$ is the embedding of neighboring spatial location $j$. The pair $(h_i, r_i)$ forms a positive pair, reflecting spatial proximity and biological similarity. To create negative examples, we generate a shuffled embedding matrix $H'$ by randomly permuting the rows of $H$, destroying spatial coherence. For each $i$, let $h'_i$ be the shuffled embedding and $r'_i$ its corresponding shuffled proxy, forming a negative pair $(h'_i, r'_i)$. Finally, we train a discriminator $\Phi$ via binary cross-entropy loss to distinguish these positive from negative pairs. The slice-specific contrastive loss is: $\mathcal{L}_{\text{SCL}}^k = -\frac{1}{n_k} \sum_{i=1}^{n_k} \left[ \log \Phi(h_{ki}, r_{ki}) + \log\big(1 - \Phi(h'_{ki}, r'_{ki})\big) \right]$ for $k = 1, 2$. The overall contrastive loss is $\mathcal{L}_{\text{SCL}} = \mathcal{L}_{\text{SCL}}^1 + \mathcal{L}_{\text{SCL}}^2$ .

### 2.4 Final Loss Function

In summary, the training objective of JADE comprises three components:

$$\mathcal{L}_{\text{SCL}} + \lambda_2 \mathcal{L}_{\text{recon}} + \lambda_3 \mathcal{L}_{\text{maintain}} + \lambda_4 \mathcal{L}_{\text{align}} + \lambda_5 \mathcal{L}_{\text{marginal}}.$$

Throughout the paper, we set $\lambda_2 = 10$, $\lambda_4 = 0.1$, $\lambda_5 = 1$. We use a data-driven method to select $\lambda_3$ from 0.2 to 2.0 depending on a calculated similarity score between slices. If two slices are similar, we use a larger $\lambda_3$ encourage information sharing; otherwise, we use a smaller $\lambda_3$ to prevent negative transfer. A comprehensive description of hyperparameter tuning procedures is provided in Appendix E. Furthermore, the contribution of each individual component in JADE is evaluated through systematic ablation studies, as detailed in Appendix F. Results show that performance deteriorates markedly when any loss term is omitted, demonstrating that both the misalignment and mismaintain losses are crucial for optimal embedding and alignment quality.

**Downstream analysis.** After training, we obtained two sets of low-dimensional embeddings. We normalized the length of each embedding vector and applied the `mclust` algorithm independently, specifying the cluster number based on prior knowledge. Following [81, 38], we set the cluster number from 5 through 7 for the DLPFC dataset, and following [20], from 16 through 17 for the axolotl brain dataset, with each cluster corresponding to a distinct cell type. We then leveraged the inferred domains to conduct differential expression analysis. For the batch-effect correction evaluation, we projected the embeddings into a 2-dimensional space using UMAP, visualized their spatial distribution, and computed the local Simpson diversity index to quantify how well the embeddings from different slices are intermingled. To assess alignment quality, we visualized and quantified the highest-probability correspondences in the alignment matrix within each ground-truth domain, explicitly marking correct and incorrect matches, and computed accuracy scores to measure alignment performance.

## 3 Fast Computation for JADE

A major challenge for SRT alignment methods is their high computational cost, scaling quadratically with the number of spatial locations. Methods such as PASTE [79] and our JADE algorithm require computing similarity matrices between all location pairs, resulting in prohibitive runtime and memory usage for high-resolution datasets. The primary computational bottleneck in JADE is the cross-attention step, which calculates pairwise similarities and scales as $O(n_1 n_2 d)$. To accelerate this, we introduce an approach that aligns at a coarser resolution using aggregated spatial units ("hyperspots"), significantly reducing computational complexity. Despite this approximation, Fast-JADE achieves performance comparable to JADE on the DLPFC dataset (see Appendix G for detailed comparison).

**Hyperspot embedding construction.** For each tissue slice, we apply $K$-means clustering to group spatial locations into fewer, coarser spatial units, resulting $m_1 \ll n_1$ and $m_2 \ll n_2$ hyperspots respectively. The number of hyperspots is set to approximately 10-20% of the original number of spatial locations for each slice. Each hyperspot embedding is computed by averaging embeddings of the spatial locations within the cluster, resulting in compact representations.

**Compute cross-attention between hyperspots.** We compute the cross-attention weights and alignment matrix at the hyperspot level. Specifically, $C^{\text{hyper}} = \text{Softmax}\left( -H_1^{\text{hyper}, T} M M^T H_2^{\text{hyper}} / \sqrt{d} \right)$ is similar to (1). Using these hyperspot-level attention weights, we then compute the alignment and maintenance losses, $\mathcal{L}_{\text{maintain}}$ and $\mathcal{L}_{\text{align}}$ at the hyperspot level.

**From hyperspot-level to spatial location-level Alignment.** After training, we transfer the learned alignment back to the full-resolution space. Specifically, we reuse the same projection head $M$ to com-

pute the fine-grained spatial location-wise attention weights: $C = \text{Softmax}\left(-H_1^T M M^T H_2/\sqrt{d}\right)$, where $H_1 \in \mathbb{R}^{n_1 \times d}$ and $H_2 \in \mathbb{R}^{n_2 \times d}$ are the original spatial location embeddings. This ensures consistency between coarse and fine levels and avoids retraining at full resolution. The resulting matrix $C$ is then passed through the Sinkhorn-Knopp algorithm to obtain an alignment matrix $\Pi$ with uniform marginals.

## 4    Experiments

We benchmarked JADE against six representative baseline models, selected for their relevance to either spatial alignment or embedding tasks in spatial transcriptomics. For alignment accuracy, we compared with PASTE [79], Seurat [57], and STAligner [81], three widely used methods that align SRT slices based on transcriptomic similarity and spatial proximity. For evaluating the learned embeddings through spatial domain detection, we included GraphST [76], STAGATE [15], and STAligner, which incorporate spatial or graph-based information to enhance domain identification.

To provide a unified assessment, we used a comprehensive alignment metric that accounts for both correctly aligned and unaligned locations (Appendix B). Embedding quality was quantified using the Adjusted Rand Index (ARI) for biological domain recovery, the integrated local inverse Simpson's index (iLISI) [54] for cross-slice integration, and UMAP visualizations for qualitative evaluation of coherence and batch correction. For biological interpretability, we conducted differential expression analysis to identify domain-specific marker genes and validated them against known anatomical annotations and literature, confirming that the learned embeddings capture biologically meaningful spatial structures and gene associations.

We also tested JADE on two additional datasets in Appendix H: the MERFISH dataset [10] and the Breast-Cancer Visium/Xenium dataset [26], previously used in SLAT [71]. As summarized in Table 11, JADE consistently outperforms or matches state-of-the-art baselines, including SLAT, STAligner and SpotScape[50], in both domain-detection accuracy (ARI) and alignment accuracy (ACC). These results demonstrate JADE's robustness and adaptability across distinct spatial transcriptomics platforms (MERFISH, Visium, Xenium) and tissue types (brain and breast).

### 4.1    Joint Spatial Alignment and Representation Learning Across Human DLPFC Slices

**Dataset description.** We evaluated our proposed method on the Human Dorsolateral Prefrontal Cortex (DLPFC) dataset generated with the 10x Visium platform [56], a standard benchmark in spatial transcriptomics. The dataset comprises 12 serial tissue sections, including four sequential slices (A–D) from each of three donors (I–III), with expression profiles of 33,538 genes measured at 47,681 spatial spots. Within each individual, slices A-B and C-D are adjacent pairs separated by 10 $\mu$m, while slices B and C have a larger separation of 300 $\mu$m. Differences across individuals are primarily attributed to batch effects arising from technical variations in sample processing, sequencing, or tissue handling. Each slice was annotated with seven spatial domains, corresponding to six cortical layers and the white matter [45]. These expert-provided annotations served as the ground truth for spatial domain detection analysis. Following preprocessing in Appendix B, the two selected slices retained about 1500 shared highly variable genes for each pair.

**Improved clustering accuracy.** Figure 2(A) provides a visual comparison of the predicted spatial domains from each method against the ground truth for both Slice A and Slice B of Sample III. Visually, JADE demonstrates the highest accuracy in recovering the annotated cortical structure, across slices A and B with clear, smooth boundaries. In contrast, other methods yield noisier and less coherent spatial domains. Quantitatively, JADE achieves median ARI values of approximately 0.61 and 0.65 for Slices A and B, for Sample III as shown in Figure 2(B). This represents a statistically significant improvement over GraphST (0.59 and 0.58), STAligner (0.56 and 0.59), and STAGATE (0.52 and 0.53). The top row of Figure 2(C) further visualizes the learned low-dimensional embeddings produced by each method using UMAP, colored by their true biological cluster annotation. JADE consistently demonstrates superior performance, yielding highly distinct and well-separated biological layers that form cohesive clusters with minimal overlap. Appendix C shows additional results for other samples and other adjacent slices.

**JADE effectively mitigates batch effects in multi-slice integration.** The bottom row of Figure 2(C) illustrates the effectiveness of batch effect removal across different methods by visualizing UMAP embeddings colored by slice identity (slice A: blue, slice B: red). Effective integration is reflected by

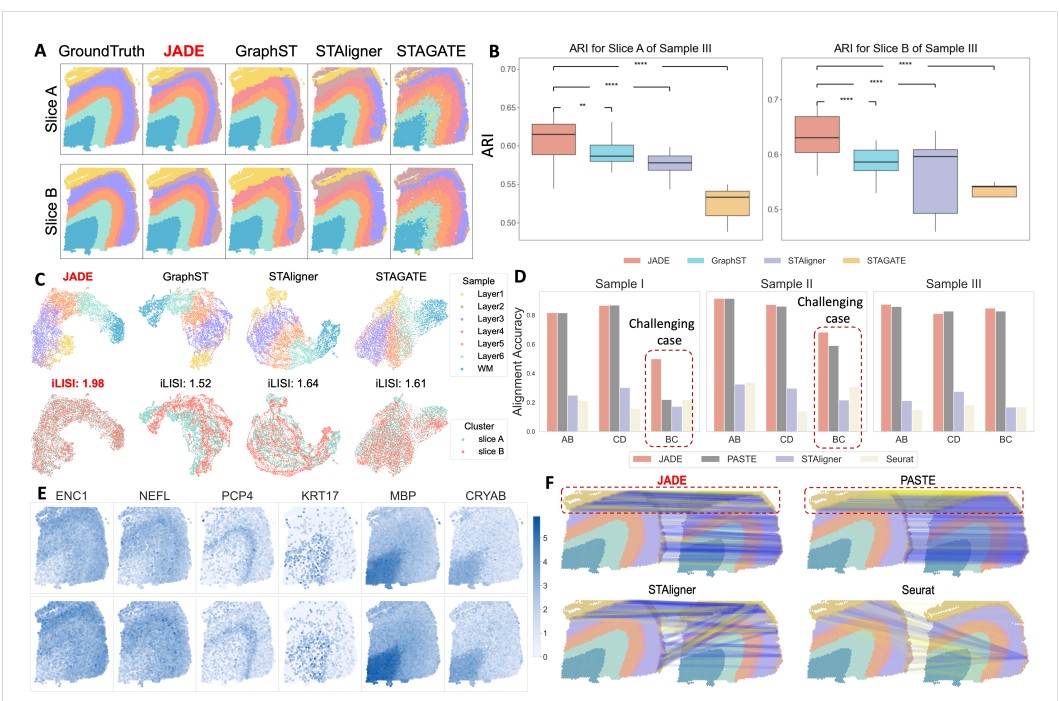

**Figure 2:** (A) Predicted spatial domains for slices A and B (Sample III) using five methods. (B) ARI for slices A and B of Sample III in DLPFC. *p*-values are calculated by Wilcoxon-rank sum test, where ** $p < 1\%$, ***$p < 0.1\%$ and **** $p < 0.01\%$ (C) UMAP of embeddings colored by predicted clusters (top) and slice (bottom). (D) Alignment accuracy across adjacent slice pairs (AB, CD, BC). (E) Spatial expression of marker genes confirms biologically relevant domain structures. (F) Visual comparison of alignment accuracy for Layer 1 in Sample III (Slice A vs. Slice B). True alignments are represented in blue; wrong ones are shown in yellow.

the intermixing of spatial locations from different slices, indicating successful alignment in a shared latent space. JADE achieves the highest inter-slice mixing, suggesting excellent batch correction performance. In contrast, GraphST, STAligner and STAGATE display limited alignment: the two slices remain largely separated within the UMAP space, suggesting poor batch removal efficacy. JADE also attained the highest iLISI score of 1.98, outperforming GraphST (1.52), STAligner (1.64), and STAGATE (1.61), confirming its superior performance in integrating SRT data across slices while correcting for batch effects.

**Superior alignment performance.** Figure 2(D) presents a comprehensive quantitative assessment of alignment accuracy across adjacent slice pairs (AB, BC, CD) for three distinct samples, comparing JADE against existing methods including PASTE, STAligner, and Seurat. Among these, the B–C pairs represent the most challenging alignment scenario due to their larger inter-slice distance ($300\mu m$), compared to the $10\mu m$ separation in A–B and C–D pairs. Across all samples and slice combinations, JADE consistently demonstrates superior alignment accuracy, particularly in challenging scenarios. While PASTE achieves comparable performance to JADE in easier settings (AB and CD pairs), it exhibits substantial performance degradation when confronted with the more challenging BC alignments. Notably, for BC pairs in samples I and II, JADE surpasses PASTE by margins exceeding 50%. STAligner shows moderate accuracy but is consistently lower than JADE and PASTE across all slice pairs. Seurat performs the poorest in all cases, with particularly low alignment accuracy in the BC pairs. These results highlight JADE's superior alignment performance, particularly in challenging integration settings. In Figure 2(F), we visualize Layer 1 alignment between Slice A and Slice B in Sample III across four methods. Blue lines indicate correct matches within the same annotated layer, while yellow lines indicate errors. JADE achieves the best outcome, with mostly correct (blue) alignments and very few incorrect (yellow) ones.

**Domain-specific differential expression gene analysis.** We performed domain-specific differential expression (DE) analysis based on the spatial domains detected by JADE and identified six representative marker genes (ENC1, NEFL, PCP4, KRT17, MBP, and CRYAB). These genes were selected for their strong domain-level enrichment and subsequently validated through literature for their well-established region-specific expression patterns in the human cortex. Figure 2(E) displays

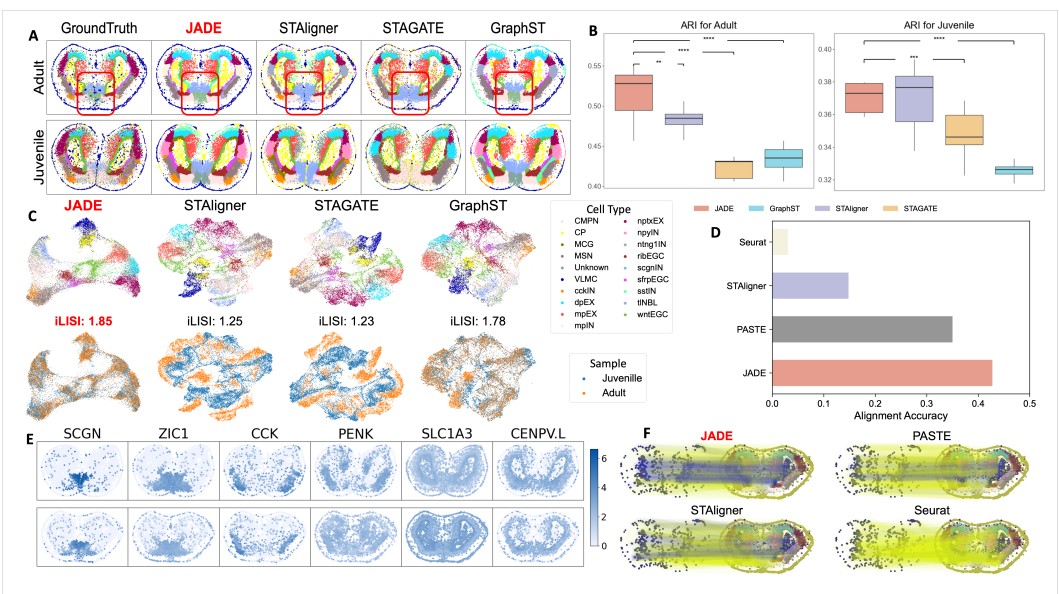

**Figure 3:** (A) Domain segmentation results for Juvenile and Adult slices comparing ground truth against four methods. (B) ARI scores for four methods. $p$-values are calculated the same as in Figure 2. (C) UMAP of embeddings colored by predicted clusters (top) and slice (bottom). (D) Alignment accuracy for Juvenile and Adult slices. (E) Spatial expression of marker genes confirms biologically relevant domain structures. (F) Visual comparison of alignment accuracy for CP cell type between the juvenile and the adult. True alignments are represented by blue lines, while wrong alignments are shown in yellow.

the spatial expression patterns of these genes across two adjacent DLPFC tissue slices (top and bottom rows). ENC1, NEFL, and PCP4 exhibit clear laminar structures consistent with neuronal populations localized to middle and deep cortical layers [45, 70, 17]. Their spatial localization is preserved across slices, demonstrating coherent biological structure. KRT17, typically associated with epithelial-like or glial cells, shows more punctate and scattered expression, particularly in the upper portions of the slices [11, 45]. MBP, a myelination-related gene, is strongly expressed in the lower regions of both slices, likely corresponding to white matter (WM), and the spatial coherence across slices further supports successful alignment [49, 45, 25]. CRYAB, a stress-response and astrocyte-associated marker, exhibits a broader expression domain with intermediate intensity [4].

Beyond expression-based baselines, we further evaluated JADE against GPSA [28], an image-informed alignment model that integrates histological features in Appendix H. As summarized in Table 10, JADE achieves higher alignment accuracy across all DLPFC slice pairs while maintaining comparable clustering quality, indicating that reliable alignment can be achieved without reliance on histology images.

### 4.2 Joint Learning of Axolotl Brain Dataset Across Different Developmental Stages

We applied our method to the task of jointly learning alignments and embeddings of SRT data across distinct developmental stages. This integration is crucial for gaining insights into the dynamic processes of cell proliferation and differentiation. However, achieving such comprehensive integration is inherently challenging due to technical batch effects and fundamental biological shifts, both of which significantly complicate the establishment of accurate underlying alignment.

**Dataset description.** We evaluated JADE on a SRT dataset of the axolotl telencephalon (a region of the brain) generated using the Stereo-seq platform [9]. This dataset captures gene expression across five developmental stages: three embryonic stages, the juvenile stage, and the adult stage. In this study, we focused our analysis on the last two, which together span a broader range of mature brain architecture. The juvenile stage and adult stage have 17 and 16 distinct cell types respectively. Among them, fourteen are shared between both stages, while the rests from the juvenile stage differentiate or transition into new cell types in the adult stage. Following preprocessing, the two selected slices retained 1,000 shared highly variable genes, with 11,698 and 8,243 spatial locations, respectively. To reduce computational time, we applied the accelerated JADE method introduced in Section 3, using 1,000 hyperspots per slice, corresponding to about 10% of the spatial locations.

**Improved clustering accuracy.** Figure 3(A) provides a visual comparison of the domains detection results of JADE and baseline methods in Adult (top row) and Juvenile (bottom row) slices. Compared to the ground truth, JADE consistently recovers structurally coherent and anatomically accurate domains in both stages. Notably, in the adult brain, JADE is the only method that successfully reconstructs the blue peripheral ring cluster, representing vascular leptomeningeal cells (VLMC) [63]. This region is either partially fragmented (in GraphST), blurred and mixed with neighboring domains (in STAGATE, STAligner). In addition, JADE accurately produces a coherent, bilaterally symmetric red region in the telencephalon core, while STAGATE yields considerable color mixing within this region, STAligner only partially recovers it, and GraphST produces an overexpanded red region, suggestive of excessive smoothing. These qualitative patterns are supported by the quantitative results in Figure 3(B). For the Adult stage, JADE achieves the highest median ARI score (approximately 0.53), significantly outperforming STAligner (0.48), STAGATE (0.42), and GraphST (0.43). For the Juvenile stage, JADE also obtains a high median ARI (approximately 0.37), demonstrating statistically significant improvement over STAGATE (0.35) and GraphST (0.32). While JADE's performance for the Juvenile stage is quantitatively comparable to STAligner's by ARI, its qualitative domain recovery remains notably more coherent and anatomically accurate as shown in Figure 3(A). The top row of Figure 3(C) further visualizes the UMAP for the embedding features, colored by the true cluster annotation. JADE produces well-separated and compact clusters, indicating a clear differentiation of spatial domains in the latent space, while other methods exhibit considerable mixing of domain colors, reflecting ambiguity in domain boundaries.

**JADE effectively mitigates batch effects in multi-slice integration.** The bottom row of Figure 3(C) compares methods via UMAP embeddings colored by slice (Adult: red, Juvenile: blue). JADE shows near-perfect mixing within structured clusters, indicating superior batch-effect removal. In contrast, STAligner, STAGATE, and GraphST display notable batch-induced separation. Quantitatively, JADE achieves the highest LISI score (1.85), surpassing GraphST (1.78), STAligner (1.25), and STAGATE (1.23), highlighting its effectiveness in integrating data across developmental stages.

**Superior alignment performance.** As shown in the bar plot in Figure 3(D), JADE achieves the highest alignment accuracy among all methods, substantially outperforming PASTE, STAligner, and Seurat. This result underscores JADE's ability to establish precise correspondences between spatial locations across slices, even under the challenging setting of cross-developmental stage alignment, where substantial morphological and transcriptional variation exists. Figure 3(F) further visualizes the alignment accuracy for cortical plate cell type between the Juvenile slice and the Adult slice, to support this finding. JADE yields the highest proportion of correct (blue) alignments, demonstrating its superior ability to preserve anatomical consistency during cross-slice registration.

**Domain-specific gene.** Figure 3(E) displays the spatial expression patterns of six canonical marker genes, SCGN, ZIC1, CCK, PENK, SLC1A3, CENPV.L, across Juvenile and Adult slices (top and bottom rows). These genes are known to exhibit region-specific expression within the axolotl brain and thus serve as internal benchmarks for spatial alignment and biological interpretability. Specifically, SCGN and ZIC1, for example, are associated with distinct neuronal populations and developmental patterning, and their laminar or domain-restricted expression patterns are preserved across developmental stages, reflecting coherent spatial organization [2, 51, 59]. CCK and PENK, which are neuropeptide-related genes, show distinct regional enrichment, highlighting the emergence of functional specialization in the maturing brain [30, 8]. SLC1A3, a glutamate transporter gene involved in astrocytic function, exhibits broad but domain-enriched expression, consistent with known glial distribution [3]. CENPV.L, associated with nuclear and centrosomal processes, displays a sharply localized expression pattern, providing a clear contrast across domains [58]. Together, these genes illustrate the biological relevance of the spatial domains identified by JADE, demonstrating its ability to uncover consistent, developmentally regulated gene expression patterns across stages of brain maturation.

# 5 Conclusions

In this paper, we address the critical yet understudied challenge of joint spatial alignment and representation learning across multi-slice SRT data. We propose JADE, a unified framework that simultaneously infers spatial correspondences and learns biologically meaningful low-dimensional embeddings. Through comprehensive evaluations on human DLPFC and axolotl brain datasets, JADE demonstrates superior performance in spatial domain detection, alignment accuracy, and batch effect correction compared to state-of-the-art methods. This study provides a robust computational tool

for integrating multi-slice SRT data, with the potential to advance 3D tissue reconstruction, cross-condition comparison, and spatially informed transcriptomic discovery. However, several limitations remain. First, while JADE performs well across a range of spatial transcriptomics platforms and tissue types, its performance may be affected by extreme sparsity, highly unbalanced slice resolutions, or limited overlap in tissue regions across slices. Second, JADE currently models pairwise slice alignment, which may limit its ability to fully reconstruct larger tissue volumes or resolve global correspondences across long serial sections. Third, the current framework assumes a static snapshot of spatial organization and does not account for temporal variation, which is increasingly relevant in developmental or regenerative contexts. Future work will extend JADE to handle fully joint alignment across multiple slices, incorporate spatiotemporal transcriptomics data, and integrate additional data modalities such as histological imaging or spatial epigenomics. We also aim to improve scalability for ultra-high-resolution datasets and explore methods to increase robustness under technical noise or sample heterogeneity. To the best of our knowledge, no potential negative impacts resulting from our work have been identified.

## 6 Acknowledgment

This work was supported by the National Science Foundation (NSF) and the National Institute of General Medical Sciences (NIGMS) under award number R01GM152814, by the National Institutes of Health (NIH) under award number R35GM160372, and by the NSF under award numbers DBI-2526948 and IIS-2500960.

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

**Input:** Slices $(A_i, X_i)$ and distance matrices $D_i$ for $i \in \{1, 2\}$; degree matrices $\mathcal{D}_i$;
        hyperparameters $\lambda_2, \ldots, \lambda_5$; epochs $T$; latent dim $d$; #Sinkhorn iterations $s$.
**Output:** Alignment $\Pi \in \mathbb{R}^{n_1 \times n_2}$; embeddings $H_i \in \mathbb{R}^{n_i \times d}$.
**Initialize** parameters $\{W_{ie}, b_{ie}, W_{id}, b_{id}, M, \Phi\}$
**for** *epoch* = 1 **to** $T$ **do**
    | // 1) Encode-Decode (GCN encoder/decoder)
    | $\tilde{A}_i \leftarrow \mathcal{D}_i^{-1/2} A_i \mathcal{D}_i^{-1/2}$;     $H_i \leftarrow \text{ReLU}(\tilde{A}_i X_i W_{ie} + b_{ie})$;
    |  $\widehat{X}_i \leftarrow \text{ReLU}(\tilde{A}_i H_i W_{id} + b_{id})$.
    | $\mathcal{L}_{\text{recon}} \leftarrow \frac{1}{n_1} \|X_1 - \widehat{X}_1\|_F^2 + \frac{1}{n_2} \|X_2 - \widehat{X}_2\|_F^2$.
    | // 2) Cross-attention & Sinkhorn for alignment
    | $S_i \leftarrow H_i M \in \mathbb{R}^{n_i \times d}$;                   // linear projection (attention)
    | $C \leftarrow \text{Softmax}_{\text{rows}}(S_1 S_2^\top / \sqrt{d}) \in \mathbb{R}^{n_1 \times n_2}$
    | $\Pi \leftarrow \text{Sinkhorn}_s(C)$;             // doubly-stochastic with marginals $(\frac{1}{n_1}, \frac{1}{n_2})$
    | $\mathcal{L}_{\text{marginal}} \leftarrow \text{KL}\left(\frac{\Pi \mathbf{1}_{n_2}}{n_2} \,\middle\|\, \frac{\mathbf{1}_{n_1}}{n_1}\right) + \text{KL}\left(\frac{\Pi^\top \mathbf{1}_{n_1}}{n_1} \,\middle\|\, \frac{\mathbf{1}_{n_2}}{n_2}\right)$.
    | // 3) Spatial-structure maintenance & embedding alignment
    | $\mathcal{L}_{\text{maintain}} \leftarrow \frac{1}{n_1} \|D_1 - n_2^2 \Pi D_2 \Pi^\top\|_F + \frac{1}{n_2} \|D_2 - n_1^2 \Pi^\top D_1 \Pi\|_F$.
    | $\mathcal{L}_{\text{align}} \leftarrow \frac{1}{n_1} \|H_1 - n_2 \Pi H_2\|_F + \frac{1}{n_2} \|H_2 - n_1 \Pi^\top H_1\|_F$.
    | // 4) Self-supervised graph contrastive loss (per slice)
    | **for** $i \in \{1, 2\}$ **do**
    |    | $r_{ij} \leftarrow \frac{1}{|N(i,j)|} \sum_{k \in N(i,j)} h_{ik}$;    // neighbor average in the spatial graph
    |    | Form positives $(h_{ij}, r_{ij})$ and negatives $(h'_{ij}, r'_{ij})$ by a row permutation of $H_i$
    |    | $\mathcal{L}_{\text{SCL}}^{(i)} \leftarrow -\frac{1}{n_i} \sum_{j=1}^{n_i} \left[\log \Phi(h_{ij}, r_{ij}) + \log(1 - \Phi(h'_{ij}, r'_{ij}))\right]$.
    | **end**
    | $\mathcal{L}_{\text{SCL}} \leftarrow \mathcal{L}_{\text{SCL}}^{(1)} + \mathcal{L}_{\text{SCL}}^{(2)}$.
    | // 5) Update
    | $\mathcal{L}_{\text{tot}} \leftarrow \mathcal{L}_{\text{recon}} + \lambda_2 \mathcal{L}_{\text{SCL}} + \lambda_3 \mathcal{L}_{\text{maintain}} + \lambda_4 \mathcal{L}_{\text{align}} + \lambda_5 \mathcal{L}_{\text{marginal}}$.
    | optimizer.step$(\nabla \mathcal{L}_{\text{tot}})$.
**end**
**return** $\Pi, H_1, H_2$



**Algorithm 1: JADE** (pairwise training loop)



## A   Pseudocode of JADE

We summarize the training loop for JADE and its accelerated variant Fast-JADE. The notation matches Sections 2 and 3: $A_i$ is the spatial graph, $X_i$ the gene-expression matrix, $\mathcal{D}_i$ the degree matrix of $A_i$, and $D_i$ the within-slice pairwise distance matrix. JADE alternates between (i) encoding/decoding to refine embeddings and (ii) alignment in the latent space via attention and Sinkhorn normalization. Fast-JADE performs alignment at a coarser "hyperspot" level for efficiency, and then recovers full-resolution correspondences using the learned projection head.

## B   Implementation details for JADE and benchmarks

**Data Preprocessing.** We followed the standardized preprocessing workflow implemented in the SCANPY package [69] to prepare the input data for our model. For each tissue slice $i$, raw gene expression counts were normalized by library size and log-transformed. Gene expression values were then scaled to unit variance across spatial locations. The top 3,000 highly variable genes (HVGs) were identified independently in each slice, and only the intersection was retained, resulting in a shared gene set of size $p < 3000$ for both datasets (around 1500 for DLPFC and around 1000 for axolotl brain dataset). This produced two input feature matrices: $X_1 \in \mathbb{R}^{n_1 \times p}$ and $X_2 \in \mathbb{R}^{n_2 \times p}$, where $n_1$ and $n_2$ denote the number of spatial locations in slices 1 and 2, respectively.

**Spatial Graph Construction.** For each tissue slice $i$, we construct an unweighted spatial graph $G_i = (V_i, E_i)$ that captures the spatial relationships among spatial locations using spatial information $S_i$. Here, $V_i$ represents the set of spatial locations, and $E_i$ consists of edges connecting neighboring locations based on their spatial proximity. To determine connectivity between spatial locations, we

**Input:** Slices $(A_i, X_i)$ and distance matrices $D_i$ for $i \in \{1, 2\}$; degree matrices $\mathcal{D}_i$; hyperspot counts $m_i$; spot-to-hyperspot assignments $N^i(l)$; hyperparameters $\lambda_2, \ldots, \lambda_5$; epochs $T$; latent dim $d$; #Sinkhorn iterations $s$.

**Output:** Alignment $\Pi \in \mathbb{R}^{n_1 \times n_2}$; embeddings $H_i \in \mathbb{R}^{n_i \times d}$.

**Initialize** parameters $\{W_{ie}, b_{ie}, W_{id}, b_{id}, M, \Phi\}$

**for** *epoch* $= 1$ **to** $T$ **do**
   // 1) Encode-Decode (same as Alg. 1)
   $\tilde{A}_i \leftarrow \mathcal{D}_i^{-1/2} A_i \mathcal{D}_i^{-1/2}; \quad H_i \leftarrow \mathrm{ReLU}(\tilde{A}_i X_i W_{ie} + b_{ie});$
   $\widehat{X}_i \leftarrow \mathrm{ReLU}(\tilde{A}_i H_i W_{id} + b_{id}).$
   $\mathcal{L}_{\mathrm{recon}} \leftarrow \frac{1}{n_1} \|X_1 - \widehat{X}_1\|_F^2 + \frac{1}{n_2} \|X_2 - \widehat{X}_2\|_F^2.$
   // 2) Hyperspot embeddings (cluster-level averaging)
   $H_i^{\mathrm{hyper}}[l] \leftarrow \frac{1}{|N^i(l)|} \sum_{j \in N^i(l)} h_j^i \quad \forall l = 1, \ldots, m_i.$
   // 3) Hyperspot-level cross-attention & Sinkhorn
   $S_i^{\mathrm{hyper}} \leftarrow H_i^{\mathrm{hyper}} M \in \mathbb{R}^{m_i \times d}$
   $C^{(0)} \leftarrow \mathrm{Softmax}_{\mathrm{rows}}\big(S_1^{\mathrm{hyper}}(S_2^{\mathrm{hyper}})^\top / \sqrt{d}\big)$
   $\Pi^{(0)} \leftarrow \mathrm{Sinkhorn}_s\big(C^{(0)}\big)$
   $\mathcal{L}_{\mathrm{marginal}} \leftarrow \mathrm{KL}\Big(\frac{\Pi^{(0)} \mathbf{1}_{m_2}}{m_2} \,\Big\|\, \frac{\mathbf{1}_{m_1}}{m_1}\Big) + \mathrm{KL}\Big(\frac{\Pi^{(0)\top} \mathbf{1}_{m_1}}{m_1} \,\Big\|\, \frac{\mathbf{1}_{m_2}}{m_2}\Big).$
   // 4) Hyperspot-level maintenance & alignment losses
   $\mathcal{L}_{\mathrm{maintain}} \leftarrow$
   $\frac{1}{m_1} \|D_1^{\mathrm{hyper}} - m_2^2 \Pi^{(0)} D_2^{\mathrm{hyper}} \Pi^{(0)\top}\|_F + \frac{1}{m_2} \|D_2^{\mathrm{hyper}} - m_1^2 \Pi^{(0)\top} D_1^{\mathrm{hyper}} \Pi^{(0)}\|_F.$
   $\mathcal{L}_{\mathrm{align}} \leftarrow \frac{1}{m_1} \|H_1^{\mathrm{hyper}} - m_2 \Pi^{(0)} H_2^{\mathrm{hyper}}\|_F + \frac{1}{m_2} \|H_2^{\mathrm{hyper}} - m_1 \Pi^{(0)\top} H_1^{\mathrm{hyper}}\|_F.$
   // 5) Graph contrastive (spot-level, identical to Alg. 1)
   Compute $\mathcal{L}_{\mathrm{SCL}}$ as in Algorithm 1.
   // 6) Update
   $\mathcal{L}_{\mathrm{tot}} \leftarrow \mathcal{L}_{\mathrm{recon}} + \lambda_2 \mathcal{L}_{\mathrm{SCL}} + \lambda_3 \mathcal{L}_{\mathrm{maintain}} + \lambda_4 \mathcal{L}_{\mathrm{align}} + \lambda_5 \mathcal{L}_{\mathrm{marginal}};$
   optimizer.step$\big(\nabla \mathcal{L}_{\mathrm{tot}}\big).$
**end**

**Recover $\Pi$:** Reuse the trained $M$ to compute $C = \mathrm{Softmax}_{\mathrm{rows}}\big((H_1 M)(H_2 M)^\top / \sqrt{d}\big)$ and then apply $\mathrm{Sinkhorn}_s$ to obtain the full-resolution $\Pi$ (as in Algorithm 1).

**return** $\Pi, H_1, H_2$

**Algorithm 2: Fast-JADE** (coarse-to-fine alignment with hyperspots)

compute the Euclidean distances between all pairs of locations using their spatial coordinates, then establish edges based on a defined neighborhood size $k$, where an edge is created between spatial location $i$ and spatial location $j$ if $j$ is among the $k$ nearest neighbors of $i$, thereby constructing a graph that captures the spatial relationships among spatial locations in the tissue slice. For all real data analyses, we set $k$ to be 3 The resulting graph is mathematically represented by an adjacency matrix $A_i = (a_{ij})$, where $a_{ij} = 1$ indicating a connection between spatial locations $i$ and $j$, and $a_{ij} = 0$ indicating no connection. After this step, we obtain two adjacency matrices of the spatial graph in two slices, $A_1 \in \mathbb{R}^{n_1 \times n_1}$ and $A_2 \in \mathbb{R}^{n_2 \times n_2}$.

**Evaluation metrics.** In this paper, we evaluate both alignment accuracy and representation learning quality across methods. To evaluate spatial alignment, we extend beyond conventional metrics that only consider the proportion of correctly aligned spatial location pairs sharing the same annotation (e.g., layer label) [79, 78]. While commonly used, these traditional metrics can be misleading: they often favor methods that selectively align "easy" spatial location pairs while ignoring harder regions, thereby inflating accuracy at the cost of coverage and interpretability. To address this limitation, we adopt an alignment accuracy metric that incorporates both aligned and unaligned spatial locations, offering a more robust and informative assessment. Specifically, let $\pi \in [0, 1]^{n_1 \times n_2}$ be the alignment matrix where $\pi_{ij} = 1$ if spatial location $i$ in slice 1 is mapped to spatial location $j$ in slice 2 (or for soft alignment methods like JADE). We normalize each column of $\pi$ to sum up to $1/n_2$, yielding a matrix $\widetilde{\pi}$ that retains soft alignments while assigning zero mass to unaligned spots. Our alignment accuracy is then defined as $\mathrm{Acc} = \sum_{i,j} \widetilde{\pi}_{ij} \mathbf{1}(l_i = l_j)$, where $l_i$ and $l_j$ denote the biological annotations (e.g., cell type, tissue layer) for spatial location $i$ and $j$, respectively. This formulation rewards biologically meaningful alignments—i.e., mappings between spots with consistent labels—while penalizing both

misalignments and omissions. When all spatial locations are perfectly aligned within the correct biological regions, Acc = 1. Finally, to ensure symmetry, we repeat the entire procedure swapping the roles of slice 1 and slice 2 and report the average of the two scores.

Beyond alignment, we evaluate the quality of the learned representations produced by each method. We use the adjusted Rand Index (ARI) to assess clustering accuracy by comparing predicted domain labels with ground-truth annotations, providing a quantitative measure of how well the representations capture underlying biological structure. To assess batch effect removal, we compute the local inverse Simpson's index (iLISI), which quantifies the degree of slice mixing in the embedding space—higher values indicate better integration across slices. We further perform qualitative assessment using Uniform Manifold Approximation and Projection (UMAP): coloring spatial locations by their true domain labels reveals biological coherence, while coloring by slice identity visually demonstrates integration quality and batch correction performance.

**Baseline methods** We benchmarked our method, JADE, against a set of six representative baseline models, selected for their relevance to either alignment or embedding tasks in spatial transcriptomics. For alignment accuracy, we evaluated against PASTE [79], Seurat [57], and STAligner [81], three widely-used methods designed to align SRT slices based on transcriptomic similarity and spatial proximity. For evaluating the learned embeddings through spatial domain detection, we benchmarked against GraphST [38], STAGATE [15], and STAligner, all of which incorporates spatial or graph-based information to enhance the detection of spatial domains. Below we outline, for each benchmark method, the key preprocessing steps, parameter settings, and clustering or alignment workflows used in our comparisons. All pipelines begin from the same raw count matrices and spatial coordinate annotations.

- **JADE.** For each pairwise alignment task, we first selected the top 3,000 highly variable genes from each slice, then took the intersection of these two sets, yielding approximately 1,500 genes for every pair in DLPFC and 1000 genes for axolotl brain dataset. We then applied log-normalization to the expression matrix, and constructed a spot-by-gene feature matrix. For the DLPFC dataset, we normalized each inter-slice distance matrix by the minimum distance between any two distinct spots within the same slice. For the axolotl brain dataset, because the juvenile slice is slightly larger physically than the adult slice, we rescaled the coordinates of the adult slice to match that of the juvenile slice. We ended up multiplying the x coordinates of the adult slice by approximately 1.3 to eliminate any alignment bias induced by scaling. The data-driven approach of selecting $\lambda_2$-$\lambda_5$ can be found in Section E. We employed a GCN with a single hidden layer to project the original expression matrix into a 64-dimensional latent space. Following Xu et al. [76], we set the neighborhood size to $k = 3$, which has been shown to yield strong performance. Before joint training, we pretrained the model to obtain an initial estimate of the alignment matrix $\Pi$, since the embeddings $H$ are essentially uninformative at the start of optimization.

$$\mathcal{L}_{\text{pretrain}} = \mathcal{L}_{\text{SCL}} + \lambda_2 \mathcal{L}_{\text{recon}} + \lambda_3 \mathcal{L}_{\text{maintain}} + \lambda_4^0 \mathcal{L}_{\text{align}}^0 + \lambda_5 \mathcal{L}_{\text{marginal}}.$$

  Similar to the Mis-Alignment loss $\mathcal{L}_{\text{align}}$, we define

$$\mathcal{L}_{\text{align}}^0 = \frac{1}{n_1} ||X_1 - n_2 \Pi X_2||_F + \frac{1}{n_2} ||X_2 - n_1 \Pi^T X_1||_F.$$

  Throughout, we fixed $\lambda_4^0 = 5.0$. We used Adam optimizer with learning rate set as 0.002, number of pretraining epochs as be 200 and number of training epochs set as 800.

- **GraphST.** Following the GraphST protocol, we used the same preprocessing steps as JADE and then assembled each spot's neighborhood by integrating within-slice 2D spatial locations. Specifically, we fixed the neighborhood size to $k = 3$ spots for each query location, which is also recommended by GraphST. All other GraphST hyperparameters were left at their package defaults. After computing the spatially informed graph, we applied the mclust Gaussian mixture model for final cluster assignment, using a smoothing radius of 20 spots in the refinement stage, which again mirroring GraphST's recommended default.

- **STAligner.** We subjected the data to our standard HVG filtering and normalization pipeline before invoking STAligner's built-in neighbor selection routine. For the human DLPFC slices, we set the cutoff radius to 8 spatial units, which yielded on average approximately 5 neighbors per spot; for the axolotl brain slices, we increased the cutoff to 50 units to

account for differences in spot density, also achieving roughly 5 neighbors per spot. All other STAligner hyperparameters remained at their defaults. Clustering on the aligned feature space was again performed with mclust, allowing direct comparability to GraphST and JADE.

- **STAGATE.** After the shared preprocessing steps, we turned to STAGATE's graph attention framework. We utilized its default neighbor selection within each slice, specifying a radius of 8 units for DLPFC (approximately 5 neighbor spots) and 50 units for the axolotl data (approximately 5 neighbor spots). This radius-based neighborhood ensures that each spot aggregates information from a consistent local context before passing through the graph attention layers. We retained STAGATE's default training hyperparameters and extracted the learned embeddings for clustering via mclust.

- **Seurat.** We applied the canonical Seurat integration workflow across all four consecutive slices of each sample in DLPFC dataset. We selected the top 3,000 variable genes for integration and computed 15 principal components on the combined expression matrix.

- **PASTE.** Finally, we ran the original PASTE algorithm with its default settings. Throughout, we set $\alpha = 0.1$ as recommended. No parameter tuning was performed beyond the defaults supplied by the PASTE package.

**Biological applications.** To assess the biological interpretability of the learned representations, we performed downstream analysis of domain-specific gene expression. For each spatial domain identified from the embeddings, we conducted differential expression analysis using the Wilcoxon rank-sum test to identify domain-enriched marker genes. These markers were then validated against known anatomical annotations and literature-curated gene lists. This analysis demonstrates the ability of our method to recover spatially organized, functionally coherent tissue structures and to reveal biologically meaningful gene–domain associations.

## C  Additional results for DLPFC

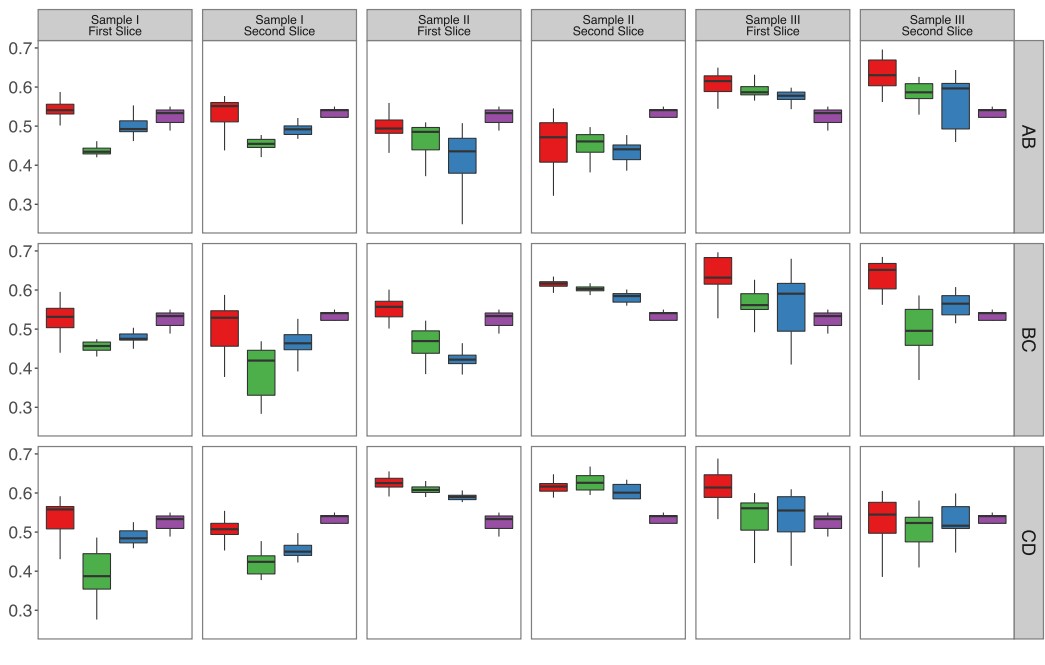

**Figure 4:** Adjusted Rand Index (ARI) results for three DLPFC samples (I–III, left to right), illustrating pairwise clustering of adjacent sections. Rows correspond to slice pairs AB, BC, and CD (top to bottom), and each boxplot summarizes ARI scores for both slices over 20 independent replicates.

Complementing Figure 2(F), Figure 4 presents the ARI results for all samples and their adjacent slice pairs in the DLPFC dataset. In nearly every comparison, JADE outperforms the other methods by

a substantial margin. Across all three DLPFC samples (I–III) and for each adjacent-slice pairing (AB, BC, and CD), JADE not only achieves the highest median ARI but also exhibits consistently tighter score distributions than the competing methods. For example, in Sample I's AB pair, JADE's median ARI is roughly 0.55, compared with about 0.45 for GraphST, 0.50 for STAligner, and 0.40 for STAGATE—an improvement of 0.05–0.15. Similarly, in Sample II's BC pairing, JADE scores near 0.62, while GraphST and STAligner both cluster around 0.60 and STAGATE lags at approximately 0.42. Moreover, JADE exhibits uniformly narrower interquartile ranges, particularly when compared to STAligner in Samples II and III and to STAGATE in Sample I. This indicates less variability and greater robustness to random initialization of JADE algorithm. Together, Figure 4 and Figure 2(E) highlight the satisfactory improvement achieved by JADE and underscore that joint alignment and embedding delivers more accurate and stable clustering across consecutive tissue sections.

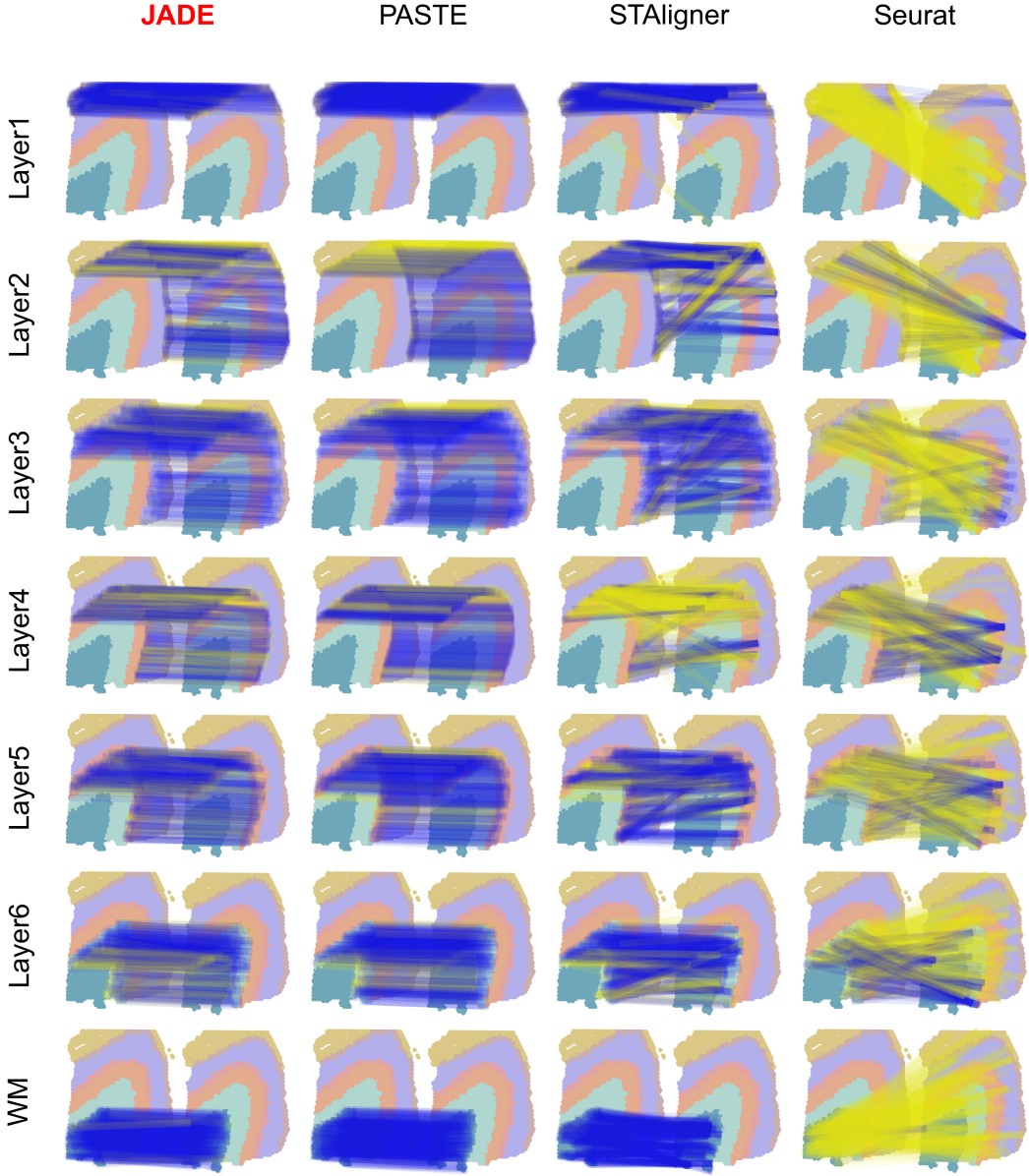

**Figure 5:** Illustration of alignment results from a fixed layer of Slice A to Slice B for Sample III, comparing four methods; blue lines denote correct correspondences, and yellow lines denote incorrect links.

Complementing Figure 2(D), Figure 5 displays the top 250 spot correspondences between a fixed layer of Slice A and Slice B for Sample III. Blue lines indicate correct matches, while yellow lines denote incorrect ones. Overall, JADE delivers superior alignment quality compared to all three other

benchmarks: it produces markedly smoother, less noisy mappings than STAligner or Seurat across all seven layers, and it achieves higher correspondence accuracy than PASTE with only a minimal trade-off in smoothness.

**Table 1:** Accuracy of top 250 alignment correspondences by domain.

| Method | Layer1 | Layer2 | Layer3 | Layer4 | Layer5 | Layer6 | WM |
|---|---|---|---|---|---|---|---|
| JADE | 0.996 | 0.636 | **0.908** | **0.776** | **0.912** | 0.752 | **1.000** |
| PASTE[79] | **1.000** | 0.584 | 0.840 | 0.724 | 0.848 | **0.836** | **1.000** |
| STAligner[81] | 0.936 | **0.648** | 0.832 | 0.252 | 0.752 | 0.740 | 0.992 |
| Seurat[57] | 0.016 | 0.084 | 0.136 | 0.296 | 0.276 | 0.344 | 0.020 |

**Table 2:** Fraction of unaligned cells per domain for each alignment method.

| Method | Layer1 | Layer2 | Layer3 | Layer4 | Layer5 | Layer6 | WM |
|---|---|---|---|---|---|---|---|
| JADE | 0.000 | 0.000 | 0.000 | 0.000 | 0.000 | 0.000 | 0.000 |
| PASTE[79] | 0.000 | 0.000 | 0.000 | 0.000 | 0.000 | 0.000 | 0.000 |
| STAligner[81] | 0.739 | 0.683 | 0.709 | 0.818 | 0.738 | 0.705 | 0.790 |
| Seurat[57] | 0.953 | 0.960 | 0.932 | 0.862 | 0.889 | 0.818 | 0.984 |

In Table 1, we report, for each spatial domain within DLPFC, the alignment accuracy achieved by the top 250 correspondence links when matching Slice A to Slice B in Sample III. Each entry shows the proportion of correctly paired spots among the first 250 alignments for that domain, providing a domain-specific assessment of alignment performance. JADE emerges as the most reliable method, achieving nearly perfect accuracy (1.00) in Layer 1 and WM and maintaining at least 0.63 accuracy in every layer. PASTE also attains perfect matches in Layer 1 and WM but shows greater fluctuation in intermediate layers (ranging from 0.584 to 0.848). STAligner also delivers competitive results in early layers, 0.936 in Layer 1 and 0.648 in Layer 2, but its performance drops markedly in deeper regions (as low as 0.252 in Layer 4), although it still nearly reaches perfection in WM (0.992). In contrast, Seurat fails to produce meaningful alignments across all domains, with accuracy scores below 0.35 in every layer (0.016–0.344). Table 2 reports, for each cortical layer (Layers 1–6) and white matter (WM), the fraction of spots left unaligned by each method. Both JADE and PASTE achieve a fully dense correspondence matrix, no spot remains unaligned in any domain. In contrast, STAligner fails to align a substantial fraction of spots (68–82% across layers, and 79% in WM), while Seurat leaves even more spots unmatched (93–96% in Layers 1–3, tapering to 81–82% in Layers 6 and 4–5, and 98% in WM). These results highlight that only JADE and PASTE guarantee fine-grained spot-spot alignment, whereas STAligner and Seurat produce large numbers of unaligned spots.

# D    Additional results for axolotl brain dataset

Complementing Figure 3(D), Figure 6 displays the top spot correspondences between a fixed layer of juvenile slice to adult slice. The umber of correspondences is determined by half of the corresponding spots. Blue lines indicate correct matches, while gray lines denote incorrect ones. Similar to DLPFC, JADE delivers superior alignment quality compared to all three other benchmarks overall.

Table 3 compares, for each common cell type between juvenile and adult slices, the accuracy of the top correspondences and the average fraction of spots left unaligned. Across all cell types, JADE attains the highest or near–highest top-link accuracy, with values ranging from 0.105 (sstIN) up to 0.766 (nptxEX). PASTE closely follows JADE, in several cases enjoying the highest accuracy for dpEX, but generally trails by 5–10 percentage points. While STAligner exceeds JADE on individual domains (e.g. scgnIN: 0.778 vs. 0.722; sfrpEGC: 0.171 vs. 0.145), JADE remains close to STAligner in some cases (e.g. scgnIN, sfrpEGC). However, STAligner's average fraction of unaligned spots is 0.724, meaning over 70 % of cells remain unaligned. Seurat performs poorly throughout, with top-link accuracies below 0.10 in every domain and an average of 0.925 unaligned spots. Taken together, these results underscore that although STAligner can rival JADE in top-ranked matches for certain cell types, its high rate of unaligned spots prevents a truly fine-grained, cell-to-cell alignment. In contrast, JADE (and PASTE) deliver both high accuracy and cell-cell alignment, ensuring no cell is left unmatched. JADE (and PASTE) combine high top-link accuracy with exhaustive alignment, leaving none of spots unaligned, making them the only methods to guarantee both precision and completeness of correspondences.

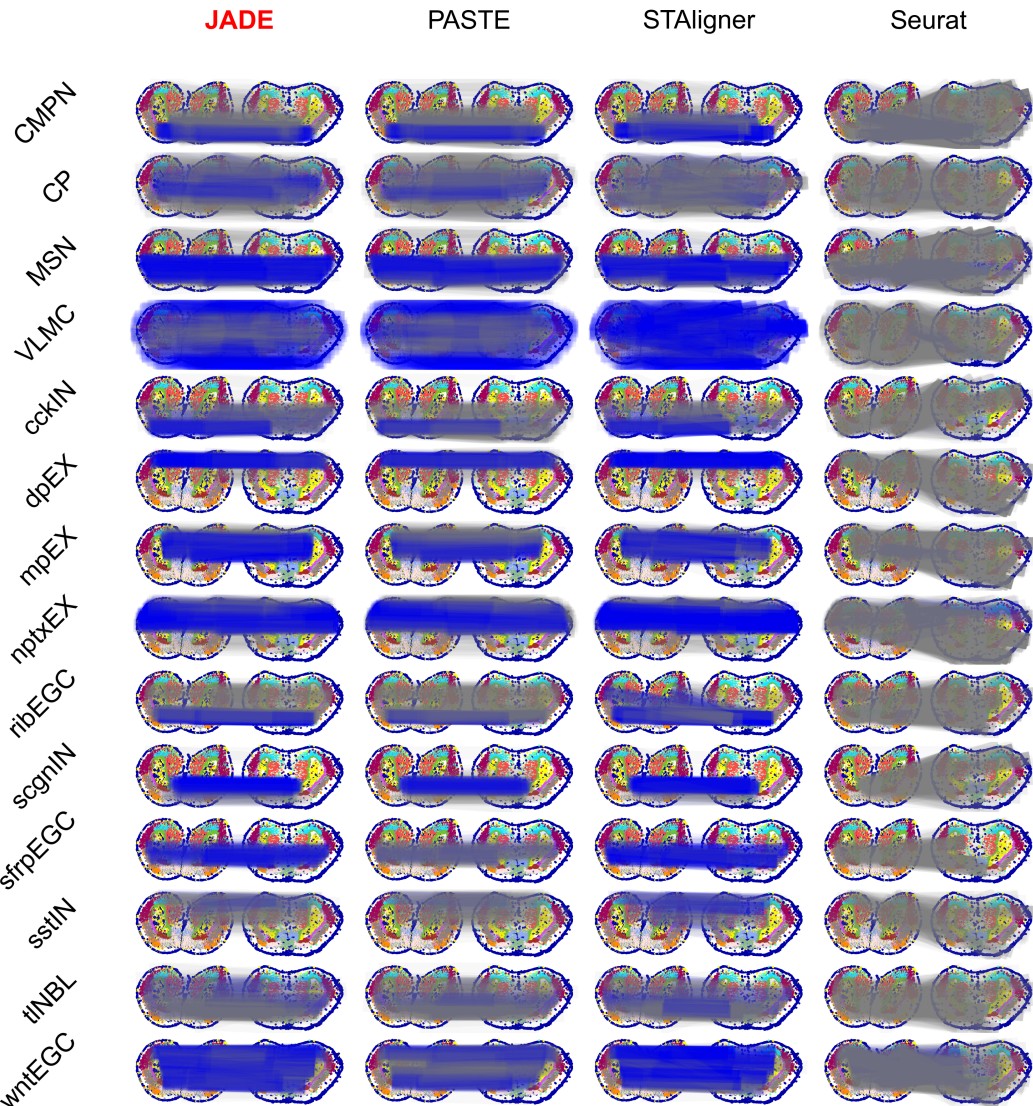

**Figure 6:** Alignment from a given cell type in juvenile slice to the adult slice, comparing four methods (JADE, PASTE, STAligner, and Seurat). Blue lines denote correct correspondences, while grey lines denote incorrect links.

**Table 3:** Alignment accuracy of the top correspondences and average fraction of unaligned spots for each common cell type between juvenile and adult slices. For each cell type, the number of correspondences equals half of its total cells.

| Domain | # cells | JADE | PASTE[79] | STAligner[81] | Seurat[57] |
|---|---|---|---|---|---|
| CMPN | 782 | 0.357 | 0.288 | **0.448** | 0.012 |
| CP | 1500 | **0.224** | 0.157 | 0.121 | 0.069 |
| MSN | 54 | 0.685 | 0.704 | **0.741** | 0.037 |
| VLMC | 523 | 0.585 | 0.539 | **0.772** | 0.011 |
| cckIN | 99 | 0.211 | 0.169 | **0.237** | 0.019 |
| dpEX | 456 | **0.809** | 0.568 | 0.721 | 0.011 |
| mpEX | 464 | **0.623** | 0.394 | 0.591 | 0.012 |
| nptxEX | 840 | **0.751** | 0.637 | 0.744 | 0.031 |
| ribEGC | 187 | 0.337 | 0.096 | 0.690 | 0.011 |
| scgnIN | 713 | 0.421 | **0.445** | 0.372 | 0.059 |
| strpEGC | 349 | 0.630 | 0.166 | **0.745** | 0.011 |
| sstIN | 294 | 0.109 | 0.075 | **0.306** | 0.003 |
| tlNBL | 257 | 0.183 | 0.105 | **0.237** | 0.000 |
| wntEGC | 509 | **0.658** | 0.326 | 0.633 | 0.069 |
| *Avg. frac. unaligned* | — | **0.000** | **0.000** | 0.724 | 0.925 |

## E  Hyperparameter tuning and sensitivity analysis

The training objective of JADE is

$$\underbrace{\mathcal{L}_{\mathrm{SCL}} + \lambda_2 \mathcal{L}_{\mathrm{recon}}}_{\text{single slice loss}} + \underbrace{\lambda_3 \mathcal{L}_{\mathrm{maintain}} + \lambda_4 \mathcal{L}_{\mathrm{align}}}_{\text{alignment loss}} + \lambda_5 \mathcal{L}_{\mathrm{marginal}}.$$

The overall loss can be decomposed into three terms: the single-slice reconstruction loss, the alignment loss, and a regularization term. Throughout, we fix $\lambda_2 = 10$ and $\lambda_5 = 1$ and choose a neighborhood size of $k = 3$. The hyperparameters $\lambda_3, \lambda_4$ govern the trade-off between enforcing accurate slice-to-slice alignment and preserving meaningful, slice-specific embeddings. Within the alignment loss, $\mathcal{L}_{\mathrm{maintain}}$ acts as a regularization term or prior belief that encodes our prior belief in the spatial coordinate similarity between the two slices. For the DLPFC dataset, we normalize each inter-slice distance matrix by the minimum distance between any two distinct spots within the same slice. We recommend using $\lambda_4 = 0.1$ and $\lambda_3 = 2.0$ by default, except in two cases—where the slices are a priori less similar—in which we set $\lambda_3 = 0.2$. To quantify slice-to-slice similarity—and thereby guide hyperparameter selection—we define the mini-max distance, which measures the average feature mismatch between corresponding spots after neighborhood smoothing. Specifically, suppose we have two slices with $n_1$ and $n_2$ spots. We first perform joint PCA and reduce the normalized gene expression matrices into low-dimensional features $Z_1 \in \mathbb{R}^{n_1 \times d}$ and $Z_2 \in \mathbb{R}^{n_2 \times d}$. Denote the adjacency matrices of these two slices by $A_1$ and $A_2$, respectively. We define the mini-max measure as follows:

$$\mathrm{Mini\text{-}max}(Z_1, Z_2) = \frac{\text{quantile}\left\{\min_{1 \le j \le n_2} \|(A_1 Z_1)_i - (A_2 Z_2)_j\| : 1 \le i \le n_1, \, 0.99\right\}}{\sqrt{\mathrm{mean}_i \|(A_1 Z_1)_i\| \times \mathrm{mean}_j \|(A_2 Z_2)_j\|}},$$

where the inner minimum is taken row-wise over the first slice's smoothed features, quantifying the distance between each spot of the first slice and its closest counterpart on the second slice. The mini-max distance is then defined as the 99%th-largest quantile among these minimal distances. Table 4 reports the mini–max distances for three samples across their adjacent slice pairs (AB, BC,

**Table 4:** Mini-max distances quantifying slice-to-slice similarity for three samples.

| Sample | AB | BC | CD |
|---|---|---|---|
| Sample I | 0.2726 | **0.5670** | 0.2609 |
| Sample II | 0.2958 | **0.4713** | 0.2764 |
| Sample III | 0.2439 | 0.2127 | 0.2368 |

CD). We note that the BC pairs in Sample I and Sample II have substantially larger distances than the rest. For these two cases, we set $\lambda_3 = 0.2$. Consequently, for all other cases, we set $\lambda_3 = 2.0$

to promote joint learning between slices. Figure 7 and Figure 8 present sensitivity analyses for the hyperparameters $\lambda_3$ and $\lambda_4$, respectively. Both the adjusted Rand index and the alignment score remain stable over a wide range of values: $\lambda_3$ from 0.5 to 2.0 times the default value; $\lambda_4$ from 0.05 to 0.25, deteriorating only when either hyperparameter is set to very low levels.

Although we fixed $\lambda_2, \lambda_5$ throughout the experiment, we extend the sensitivity analysis to $\lambda_2$ and $\lambda_5$ in Table 5 to further demonstrate the flexibility of our algorithm.

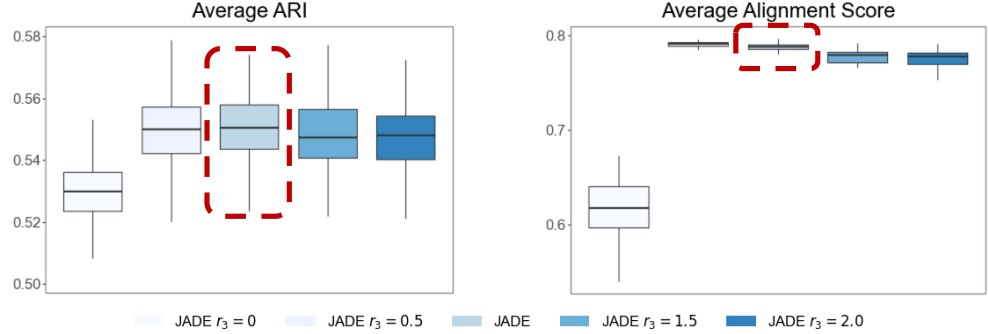

**Figure 7:** Sensitivity analysis: Varying $\lambda_3$, where $r_3$ is the magnifier of $\lambda_3$ against JADE. Each boxplot summarizes the outcomes of 100 independent replications.

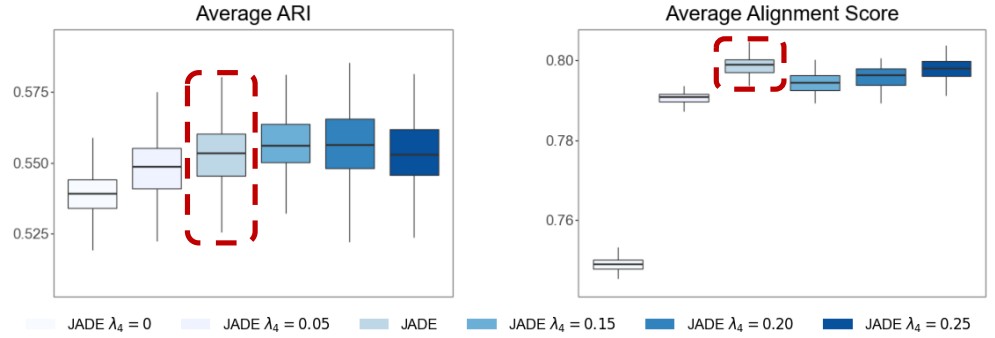

**Figure 8:** Sensitivity analysis: Varying $\lambda_4$. Each boxplot summarizes the outcomes of 100 independent replications.

## F Ablation study

**Table 5:** Sensitivity analysis of $\lambda_2$ and $\lambda_5$ for JADE.

| Metric | Default | $0.5 \times \lambda_2$ | $2.0 \times \lambda_2$ | $2.5 \times \lambda_2$ | $0.5 \times \lambda_5$ | $2.0 \times \lambda_5$ | $2.5 \times \lambda_5$ |
|---|---|---|---|---|---|---|---|
| ARI | 0.550 | 0.550 | 0.564 | 0.564 | 0.547 | 0.545 | 0.552 |
| Alignment ACC | 0.788 | 0.797 | 0.790 | 0.788 | 0.801 | 0.773 | 0.787 |

Figure 9 compares JADE to versions with no mismaintain loss ($\lambda_3 = 0$) and no misalignment loss ($\lambda_4 = 0$). Results show that performance deteriorates markedly when either loss term is omitted, demonstrating that both the misalignment and mismaintain losses are crucial for optimal embedding and alignment quality. Setting $\lambda_4 = 0$ causes the ARI to drop from 0.55 to 0.53 (a 4% decrease) and the alignment score to fall from 0.8 to 0.7 (over a 10% decrease), underscoring the critical role of the misalignment loss. Setting $\lambda_3 = 0$ causes the ARI to drop from 0.55 to 0.525 (a 5% decrease) and the alignment score to fall from 0.8 to 0.62 (about 20% decrease), underscoring the critical role of the mismaintain loss.

## G Scalability of Fast-JADE

Table 6 and 7 present the runtime and average ARI for JADE and Fast-JADE, the accelerated version of JADE using hyperspots as introduced in Section 3. We see that as the runtime reduce

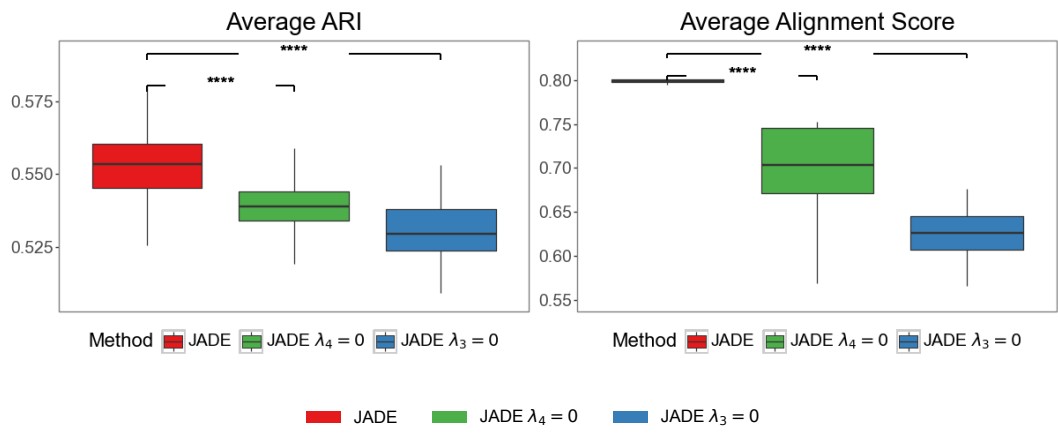

**Figure 9:** Ablation study: each boxplot summarizes the outcomes of 100 independent replications. We calculate the $p$-value using the Wilcoxon-rank sum test. **** stands for $p$-value lower than 0.01%

**Table 6:** Runtime comparison of JADE and Fast-JADE (relative run time).

| Method | DLPFC | Axolotl Brain |
|---|---|---|
| JADE | 1.000 | 1.000 |
| Fast-JADE (with 2000 hyperspots) | 0.504 | 0.640 |
| Fast-JADE (with 1000 hyperspots) | 0.200 | 0.333 |

significantly when using Fast-JADE instead of the standard version of JADE while the ARI remains almost the same. Table 8 presents the GPU runtime per epoch for Fast-JADE. As shown in the table, Fast-JADE demonstrates approximately linear scaling with respect to the data size, maintaining efficient performance even on large-scale data.

# H  Discussion

**Direct alignment against transitive alignment**. We conducted additional experiments on the DLPFC dataset (Sample III) using slices A, B, and C. Specifically, we first computed pairwise alignments $\Pi_{AB}$ (between A and B) and $\Pi_{BC}$ (between B and C) using JADE, and then derived a transitive alignment $\Pi_{AC} = \Pi_{AB} \times \Pi_{BC}$, followed by Sinkhorn normalization to enforce the doubly stochastic property. We compared this transitive $\Pi_{AC}$ with the direct alignment obtained by running JADE on slices A and C. The alignment accuracy for the direct A–C alignment was 0.767, whereas the transitive alignment via A–B–C achieved an accuracy of 0.749. These results suggest that although transitive alignment using JADE is feasible, it accumulates intermediate alignment noise, resulting in slightly lower accuracy than direct pairwise alignment. This finding highlights an important point: JADE's pairwise design supports flexible alignment across arbitrary slice pairs and enables indirect mapping when necessary, but direct alignment remains the preferred strategy due to its superior accuracy.

**Performance under unbalanced number of spots in two slices**. We conducted an experiment on the DLPFC dataset (Sample III, slices A and B), where we randomly masked a subset of spots from slice A to simulate scenarios with unequal spot counts. Table 9 reports the results obtained by varying the proportion of removed spots (10%, 25%, and 50%) in the DLPFC data.

**Table 7:** Average ARI comparison of JADE and Fast-JADE in DLPFC.

| Method | DLPFC |
|---|---|
| JADE | 0.551 (0.002) |
| Fast-JADE (with 1000 hyperspots) | 0.536 (0.001) |

**Table 8:** GPU runtime for Fast-JADE (milliseconds per epoch).

| Number of spots per slice | 10,000 | 20,000 | 50,000 | 100,000 |
|---|---|---|---|---|
| Runtime (ms/epoch) | 14.6 | 32.3 | 84.8 | 177 |

Across all tested levels of spot removal (up to 50%), the Adjusted Rand Index (ARI) for slices A and B, overall alignment accuracy (ACC), and batch-correction metric (iLISI) remain highly stable, showing only minimal fluctuations. This robustness demonstrates that JADE effectively handles situations with substantially different spot counts between slices, making it practical for real-world applications where tissue sections vary in size.

**Table 9:** Performance under unequal number of spots on the DLPFC dataset.

| % Spots Removed | ARI (Slice A) | ARI (Slice B) | ACC | iLISI |
|---|---|---|---|---|
| 0% | 0.62 | 0.65 | 0.83 | 1.98 |
| 10% | 0.60 | 0.66 | 0.81 | 1.97 |
| 25% | 0.61 | 0.65 | 0.82 | 1.98 |
| 50% | 0.62 | 0.67 | 0.82 | 1.98 |

**Comparison with image-based alignment methods**. To evaluate JADE against image-based alignment methods, we conducted an experiment with GPSA [28], the only method in the benchmarking study by Hu et al. [24] that integrates both gene expression and histology features. As shown in Table 10, JADE consistently outperforms GPSA in alignment accuracy across all slice combinations in the DLPFC dataset.

While incorporating histological images can, in principle, improve alignment, their effectiveness depends heavily on image quality, resolution, and cross-sectional consistency. In practice, H&E images may be noisy, misaligned, or inconsistently stained. Moreover, these images are often obtained from adjacent rather than identical sections, limiting their spatial correspondence with transcriptomic profiles. Such discrepancies can introduce spurious or noisy signals, particularly when histological structures do not clearly delineate molecular domains.

By contrast, JADE relies on gene expression and spatial location information, which are more consistently measured across spatial transcriptomics platforms. This design enables JADE to be broadly applicable to technologies such as the Stereo-seq platform (as shown in Figure 3), where histological imaging is unavailable, and to remain robust when image quality is variable or inconsistent.

Importantly, JADE's framework is modular and can be extended to incorporate image-derived information in future work. For example, histological features could be integrated by modulating the spatial graph (e.g., assigning image informed weights to graph edges) or by combining image embeddings with expression-based representations. These extensions could further enhance JADE's utility in contexts where high-quality image data are available, while preserving its core advantage of joint alignment and representation learning. Unlike alignment-only methods such as GPSA or PASTE, JADE uniquely supports both spatial alignment and shared low-dimensional embedding, enabling a broader range of downstream analyses, including clustering, visualization, and trajectory inference.

**Table 10:** Alignment accuracy (ACC) comparison between JADE and GPSA on the DLPFC dataset (Samples I–III; slices A–D).

| Method | Sample I | | | Sample II | | | Sample III | | | Average |
|---|---|---|---|---|---|---|---|---|---|---|
| | AB | BC | CD | AB | BC | CD | AB | BC | CD | |
| JADE | **0.76** | **0.54** | **0.81** | **0.88** | **0.76** | **0.84** | **0.83** | **0.82** | **0.79** | **0.78** |
| GPSA[28] | 0.19 | 0.20 | 0.25 | 0.42 | 0.34 | 0.28 | 0.18 | 0.17 | 0.17 | 0.24 |

**Comparison with new benchmarks**. To further evaluate the generalizability and robustness of JADE, we conducted additional experiments on two diverse and challenging datasets: the MERFISH dataset [10] and the breast cancer Visium/Xenium dataset [26], previously used in SLAT [71]. In the latter, one slice was generated using Visium (approximately 3,500 spots and over 15,000 genes) and the other using Xenium (over 140,000 spots but only about 300 genes). As shown in Table 11, JADE consistently outperforms or matches existing methods, SLAT and STAligner, in both domain detection

accuracy (Adjusted Rand Index, ARI) and alignment accuracy (ACC). These results demonstrate JADE's robustness across distinct spatial transcriptomics platforms (MERFISH, Visium, Xenium) and tissue types (brain and breast).

We note an important caveat regarding the breast cancer dataset. This dataset does not provide manually curated one-to-one ground-truth correspondences between the Visium and Xenium slices. The Visium slice includes coarse region-level labels (e.g., *immune*, *invasive*), whereas the Xenium slice contains more fine-grained cell-type annotations (e.g., *CD8$^+$ T Cells*, *Invasive Tumor*). To enable quantitative evaluation, we manually harmonized these label sets based on biological correspondence and naming conventions. Although this approximation enables consistent comparison across methods, it introduces some ambiguity into the label-based evaluation. The reported alignment accuracy (ACC) should therefore be interpreted with caution, as it may partially reflect label mismatches rather than true misalignment. A more definitive assessment would require expert-annotated alignments, which are currently unavailable for this dataset.

**Table 11:** Comparison between JADE (Fast-JADE with $m_1 = m_2 = 1000$), SLAT, STAligner, and SpotScape on the MERFISH and Breast Cancer datasets. Results are averaged over 20 runs.

| Method | MERFISH[10] | | | Breast Cancer[26] | | |
|---|---|---|---|---|---|---|
| | ARI-1 | ARI-2 | ACC | ARI-1 | ARI-2 | ACC |
| JADE | **0.504** | **0.538** | **0.706** | **0.433** | **0.230** | **0.336** |
| SLAT[71] | 0.224 | 0.331 | 0.386 | 0.420 | 0.197 | 0.294 |
| STAligner[81] | 0.371 | 0.487 | 0.535 | 0.359 | 0.186 | 0.089 |
| SpotScape[50] | 0.346 | 0.265 | 0.690 | 0.277 | 0.179 | 0.312 |

