# OpenReview forum: "JADE: Joint Alignment and Deep Embedding for Multi-Slice Spatial Transcriptomics"
_NeurIPS.cc/2025/Conference — NeurIPS 2025 poster_

### Official Review · Reviewer_c7f2 · 2025-07-01

**Clarity:** 3
**Significance:** 2
**Originality:** 1
**Rating:** 4
**Confidence:** 5

**Summary:**

To address the limitations of existing methods that focus solely on either spatial alignment or representation learning, the authors propose JADE, a framework that jointly learns spatial location-wise alignments and shared low-dimensional embeddings across tissue slices. Specifically, JADE computes a cross-slice alignment matrix using the Sinkhorn-Knopp algorithm applied to cross-attention weights, and refines it using a combination of spatial structure preservation loss and embedding alignment loss. In parallel, the model learns low-dimensional embeddings via graph contrastive learning, following a strategy similar to that used in GraphST.

**Questions:**

1) Please respond to the comments in the 'Weaknesses' section.

2) In highly heterogeneous settings, such as cancer tissue, there is concern that the assumption behind the embedding alignment loss—i.e., that aligned embeddings should be close to each other—may not hold. How does the proposed embedding alignment loss account for such heterogeneity? Please provide a detailed discussion and support it with experiments on a heterogeneous dataset, such as the Breast Cancer data used in prior works (e.g., SLAT).

3) The authors state that the fast version of JADE, which utilizes hyperspots, can reduce computational time. However, it remains unclear whether this approach can scale effectively to very large datasets, which are increasingly common and may contain over 100,000 spots. Please clarify the scalability of Fast-JADE by providing a runtime analysis on such large-scale data.

**Ethical Concerns:**

["NO or VERY MINOR ethics concerns only"]

**Final Justification:**

I have concerns regarding the novelty of the proposed method; however, the paper demonstrates its effectiveness across multiple datasets and tasks, and the authors added important experiments during the rebuttal period.

**Limitations:**

yes

**Quality:**

2

**Strengths And Weaknesses:**

* Strengths
The authors propose a unified framework, JADE, that goes beyond single-task specific methods by performing multiple tasks through jointly learned alignments and representations. Notably, the inclusion of domain-specific gene (marker gene) detection demonstrates that the learned representations can carry biologically meaningful insights. This highlights the potential of JADE for downstream biological interpretation.

* Weaknesses
1) The main motivation of the paper is that existing methods do not jointly learn spatial location-wise alignments and shared low-dimensional embeddings across tissue slices. However, recent methods such as SLAT [1] and CAST [2] already address both aspects. The key difference lies in their two-stage pipelines (first learn embeddings, then perform alignment), whereas JADE integrates both steps into a unified process. To justify the claimed advantage of this one-step design, the authors should include direct comparisons with SLAT and CAST, both in methodology and empirical performance.

2) There are concerns regarding the novelty of the proposed framework. The components of the proposed method closely resemble existing work. The contrastive learning strategy for representation learning is similar to that used in GraphST, and the alignment component shares conceptual similarities with PASTE, differing mainly in that JADE uses learned embeddings rather than raw gene expression profiles. The authors should clarify the methodological distinctions and contributions of JADE relative to these prior works.

3) The experiments are limited to the DLPFC and Axolotl Brain datasets. However, spatial transcriptomics data vary significantly depending on the platform (e.g., Visium, MERFISH, Slide-seq) and tissue type. To demonstrate robustness and generalizability, the authors should evaluate JADE on additional datasets, such as the MERFISH dataset used in STAligner or the breast cancer data used in SLAT.

[1] Spatial-linked alignment tool (SLAT) for aligning heterogenous slices. Nature communications
[2] Probabilistic embedding, clustering, and alignment for integrating spatial transcriptomics data with PRECAST. Nature communications

---

> ### Author Rebuttal · Authors · 2025-07-31
>
> We thank the reviewer for constructive comments. We will add the following discussions, additional experiments, and references to the revised manuscript.
>
> >  ***Comment:** direct comparisons with SLAT and PRECAST (CAST).*
>
> **Response:** Thank you for this valuable comment.
>
> **Methodological Distinction:** SLAT and PRECAST employ a two-stage approach, first learning low-dimensional embeddings independently per slice, then aligning based on these, which can yield suboptimal embeddings as they cannot adapt to alignment insights, and vice versa. **In contrast**, JADE simultaneously optimizes alignment and embeddings through a unified roundtrip process. This joint optimization is advantageous, as alignment and embedding are **inherently interdependent**: accurate alignments rely on embeddings capturing shared transcriptomic patterns, while robust embeddings benefit from alignments resolving spatial correspondences and reducing distortions or batch effects.
>
> Beyond architectural differences, SLAT and PRECAST have methodological limitations: (1) SLAT’s initial SVD-based decomposition assumes a dominant low-rank structure, limiting its flexibility in capturing complex or heterogeneous spatial domains. In contrast, JADE’s end-to-end embedding learning with alignment adapts to diverse biological variations without such assumptions. (2) The embedding of PRECAST highly depends on pre-specifying spatial domain numbers (with BIC for selection); JADE avoids this by learning flexible embeddings compatible with any downstream clustering or domain detection method.
>
> **Empirical Comparison:** As suggested, we implemented experiments to directly compare JADE, SLAT, and PRECAST on two benchmark datasets: DLPFC (sample III, slice A and B) and Axolotl Brain datasets. Table R4 shows that JADE consistently achieves both better domain detection accuracy (higher ARI) and alignment accuracy (higher ACC). This highlights the advantage of jointly training alignment and embeddings.
>
> Table R4: Comparison between JADE, SLAT, and PRECAST on DLPFC  and Axolotl Brain datasets. Bold values indicate the best performance for each metric.
>
> | **Dataset** | **ARI (Slice A)** |   |  | **ARI (Slice B)** |   |  | **ACC** |   |  |
> | ----------- | :---------------: | :---: | :--: | :---------------: | :---: | :--: | :-----: | :---: | :--: |
> |             |        JADE       |  SLAT | PRECAST |        JADE       |  SLAT | PRECAST |   JADE  |  SLAT | PRECAST |
> | DLPFC       |       **0.551**       | 0.323 |  0.400    |       **0.550**       | 0.329 |   0.440   |  **0.799**  | 0.780 |  0.399    |
> | Axolotl     |       **0.375**       | 0.367 |   0.361   |       **0.532**       | 0.466 |  0.416    |  **0.421**  | 0.273 |  0.401    |
>
> >  ***Comment:** methodological distinctions and contributions of JADE relative to these prior works.*
>
> **Response:** Thank you for this thoughtful comment and the opportunity to clarify the methodological contributions of JADE in relation to prior work, particularly GraphST and PASTE. While JADE draws inspiration from these methods, its **core innovation** lies in unifying these into a joint, iterative framework that mutually refines alignments and embeddings, addressing the limitations of prior two-stage or isolated approaches. Below, we outline the key distinctions.
>
> 1. **Methodological distinction from GraphST:** GraphST does not explicitly model spatial location-level alignments across slices, treating integration as a secondary step without iterative feedback. JADE builds on contrastive learning but integrates it within a roundtrip framework, where embeddings are refined alongside alignments using attention-based OT, ensuring spatial correspondences directly inform feature learning (and vice versa). This joint optimization prevents inconsistencies, such as misaligned homologous regions leading to noisy embeddings, which GraphST may overlook.
> 2. **Methodological distinction from PASTE:** PASTE uses fused Gromov-Wasserstein OT on raw profiles for alignment, excelling in 3D reconstruction but not learning embeddings, limiting downstream utility. JADE applies OT to learned embeddings with attention weighting to focus on relevant features while suppressing noise, and its roundtrip iteration allows joint optimization of alignments and embeddings.
> 3. **Key Innovations and Contributions of JADE:** By jointly optimizing alignment and representation learning in a shared latent space, it creates a feedback loop where alignments guide embeddings to better capture cross-slice consistencies, and embeddings enhance alignments by providing denoised, biologically meaningful features. This is particularly beneficial in noisy SRT data, where raw profiles (as in PASTE) can be sparse and batch-affected, or independent embeddings (as in GraphST) may fail to resolve distortions. To our knowledge, **JADE is the first framework to jointly optimize for spatial alignment and shared low-dimensional embedding in an end-to-end manner.** Empirically, the above Table R4 further illsutrate our advanatge when compared  with two-stages methods: SLAT and PRECAST.
>
> >  ***Comment:** The experiments are limited to the DLPFC and Axolotl Brain datasets.*
>
> **Response:** Thank you for your thoughtful suggestion. Following your advice, we have expanded our experimental evaluation to include two additional and diverse datasets: the MERFISH dataset and the breast cancer Visium/Xenium dataset (as used in SLAT), where one slice was generated using Visium (~3,500 spots with >15,000 genes) and the other using Xenium (>140,000 spots but ~300 genes). Results from these datasets are presented in Table R5, showing that JADE consistently outperforms or matches prior methods in both domain detection (ARI) and alignment accuracy (ACC). These findings further support JADE’s robustness across different platforms (MERFISH, Visium, Xenium) and tissue types (brain and breast).
>
> We would like to note one important caveat regarding the breast cancer dataset: it lacks manually curated one-to-one ground truth correspondences between the Visium and Xenium slices. The Visium data provides coarse region-level labels (e.g., "immune", "invasive"), while the Xenium data [1] provides fine-grained cell type labels (e.g., "CD8+_T_Cells", "Invasive_Tumor"). For evaluation, we manually harmonized labels based on biological correspondence and naming conventions. However, this mismatch introduces ambiguity into the label-based evaluation metric. Establishing accurate ground truth alignment would require expert annotation by biologists, which is not currently available for this dataset. As a result, the reported alignment accuracy (ACC) should be interpreted with caution, as it may be influenced by label bias rather than true misalignment. Nevertheless, the evaluation provides a consistent basis for comparison, as all methods are assessed using the same approximate label mapping.
>
> Table R5: Comparison between JADE (Using Fast-JADE with $m_1=m_2=1000$), SLAT, and STAligner on MERFISH  and Breast Cancer datasets. Bold values indicate the best performance for each metric.
>
> | **Dataset**         | **ARI (Slice 1)** |      |      | **ARI (Slice 2)** |      |      | **ACC** |      |      |
> | ------------------- | :---------------: | :--: | :--: | :---------------: | :--: | :--: | :-----: | :--: | :--: |
> |                     |        JADE       | SLAT | STAligner |        JADE       | SLAT | STAligner |   JADE  | SLAT | STAligner |
> | MERFISH  |        **0.504**           |  0.224    |  0.371    |     **0.538**              |  0.331    |  0.487   |   **0.706**      | 0.386     |  0.535   |
> | Breast Cancer       |   **0.433**                |  0.420    |   0.359   |   **0.230**                |  0.197    |   0.186   |   **0.336**      |  0.294    |  0.089    |
>
> [1] Janesick, A., Shelansky, R., & Sarah E. B. T. (2023). High resolution mapping of the tumor microenvironment using integrated single-cell, spatial and in situ analysis. Nature Communications, 14, 8353.
>
> >  ***Comment:** experiments on a heterogeneous dataset, such as the Breast Cancer data used in prior works (e.g., SLAT).*
>
> Thank you for this constructive comment. JADE addresses cross-slice heterogeneity by combining two key strategies: (1) it employs seperate encoders and decoders for each slice, allowing the model to capture platform-specific variations such as gene coverage, resolution, and batch-specific scaling effects. (2) Embeddings from both slices are mapped into a shared latent spcae, and alignment is enforces via the misalignment loss, which encourages biologically meaningful spatial correspondence across datasets. Moreover, in settings such as the breast cancer study—where one slice is generated by Visium (~3500 spots but >15,000 genes) and the other by Xenium (>140,000 spots but only ~300 genes)—Fast-JADE clusters spots into hyperspots and aligns the slices at the hyperspot level, thereby mitigating heterogeneity. As demonstrated in the previous results, Fast-JADE outperforms SLAT and STAligner, highlighting its potential for heterogeneous datasets.
>
>
> >  ***Comment:** Please clarify the scalability of Fast-JADE by providing a runtime analysis on large-scale data.*
>
> Thank you for the insightful comment. To assess runtime performance, we conducted GPU-based runtime profiling using varying numbers of spatial locations while fixing the number of hyperspots per slice to \($m_1$ = $m_2$ = 1000\). As shown in the table below, Fast-JADE demonstrates linear scaling behavior with respect to input size, maintaining efficient performance even on large-scale data:
>
> | Number of spots                   | 10,000 | 20,000 | 50,000 | 100,000 |
> |:------------------------:|:------:|:------:|:------:|:-------:|
> | Runtime (ms/epoch)       |   14.6     |   32.3     |   84.8   |    177     |
>
> In future work, we plan to incorporate mini-batch training and sparse OT approximations to further improve scalability for ultra-large datasets.

---

> > ### Comment · Reviewer_c7f2 · 2025-08-05
> >
> > Thank you for your time in addressing my reviews.
> >
> > I understand that your proposed framework is methodologically distinct from SLAT, PRECAST, and GraphST in terms of the inherently interdependent relationship between alignment and embeddings, and also distinct from PASTE in terms of leveraging Optimal Transport (OT) to learned embeddings.
> >
> > While I still believe that the novelty of this paper is somewhat limited compared to existing works, I acknowledge that the paper presents promising results in both clustering and alignment across multiple datasets during this rebuttal period.
> >
> > However, I have one remaining concern: your model performs well in scenarios where two slices have well-aligned coordinates (e.g., the A-B and C-D slices in the DLPFC data), but seems to struggle when the coordinates are not well aligned, such as with the B-C slice in DLPFC. Could you clarify why the model behaves differently in these cases?
> >
> > Finally, I recommend conducting a sensitivity analysis regarding the balance parameter λ in the loss function, as robustness to this parameter is critical for practical performance in unsupervised scenarios. Additionally, I suggest adding SpotScape [1] as an additional baseline, as it is a state-of-the-art method that addresses similar tasks. If time constraints during the rebuttal period prevent this, these changes can be incorporated in the final version.
> >
> > [1] [ICML 2025] Global Context-aware Representation Learning for Spatially Resolved Transcriptomics

---

> > > ### Author Response · Authors · 2025-08-07
> > >
> > > >  ***Comment:** Clarify why the model behaves differently in Slice BC.*
> > >
> > > Thank you for the constuctive comment.  We would like to clarify and justify the observed results as follows:
> > >
> > > **1. Lower Alignment Accuracy for the BC Pair is a Common Challenge**. First, the reduced performance on the BC pair in Sample I and Sample II is not unique to JADE; methods including PASTE and STAligner also show lower accuracy for BC than for adjacent AB and CD pairs (Figure 2D). This reflects inherent difficulties in aligning more distant or heterogeneous slices.
> > >
> > > **2. Biological and Technical Factors Make BC Pair Challenging**. Two main factors make the BC pair particularly challenging:
> > >
> > > * **Increased Physical Separation:** The BC pair is separated by 300 μm, in contrast to the 10 μm separation of the AB and CD pairs. This larger distance leads to greater anatomical and transcriptomic divergence, with possible changes in cell composition, layer structure, and tissue morphology, making accurate alignment more difficult.
> > > * **Technical Artifacts:** Tissue sections separated by larger distances (e.g., BC pair) are more prone to artifacts such as deformation, rotation, and misregistration. In Sample I and Sample II, these effects are more noticeable in the BC pair and can further reduce spatial correspondence between slices.
> > >
> > > **3. JADE Remains Robust and Outperforms Other Methods on the BC Pair**. Despite the increased complexity, JADE consistently outperforms existing methods on the BC pair (Figure 2D). This is due to JADE’s joint optimization of alignment and embedding, which leverages a denoised, biologically informed latent space. Unlike methods that rely primarily on either spatial proximity or expression similarity, JADE’s integrative approach better captures meaningful correspondences, even across distant and structurally variable slices.
> > >
> > > >  ***Comment:** Sensitivity analysis regarding the balance parameter.*
> > >
> > > Thank you for the thoughtful suggestion. We agree that robustness to hyperparameter settings is important for unsupervised methods. As noted in Appendix D, we previously included sensitivity analyses for $\lambda_3$ and $\lambda_4$; In response to your comment, we have now extended the analysis to include $\lambda_2$ and $\lambda_5$. JADE’s performance remains consistent in both alignment and clustering accuracy when $\lambda_2$-$\lambda_5$ are scaled to **0.5x, 1.0x, 2.0x, and 2.5x** their default values. The full results are summarized below:
> > >
> > > | Metric            | Default | 0.5× λ₂ | 2.0× λ₂ | 2.5× λ₂ | 0.5× λ₃ | 2.0× λ₃ | 2.5× λ₃ | 0.5× λ₄ | 2.0× λ₄ | 2.5× λ₄ | 0.5× λ₅ | 2.0× λ₅ | 2.5× λ₅ |
> > > |:-----------------:|:---------------:|:-------:|:-------:|:-------:|:-------:|:-------:|:-------:|:-------:|:-------:|:-------:|:-------:|:-------:|:-------:|
> > > | **ARI**           |      0.550      |  0.550  |  0.564  |  0.564  |  0.552  |  0.545  |  0.545  |  0.546  |  0.556  |  0.553  |  0.547  |  0.545  |  0.552  |
> > > | **Alignment ACC** |      0.788      |  0.797  |  0.790  |  0.788  |  0.788  |  0.776  |  0.773  |  0.790  |  0.796  |  0.797  |  0.801  |  0.773  |  0.787  |
> > >
> > >
> > > We hope this additional analysis addresses your concern and confirms the robustness of JADE under varying hyperparameter settings.
> > >
> > > >  ***Comment:** Adding SpotScape as baseline.*
> > >
> > > Thank you for your insightful suggestion. We evaluated JADE against SpotScape on multiple datasets (see Table A below) and found that JADE consistently outperforms SpotScape in both ARI and alignment accuracy.
> > >
> > > SpotScape is a graph-based method for spatial domain detection and multi-slice integration. Similar to STAligner, it is an integration-only approach: embeddings are learned independently for each slice, and alignment is performed afterward. This decoupled design can lead to ambiguous correspondences, with some spots unmatched or multiply matched. In contrast, JADE jointly learns alignment and embedding, producing more coherent and biologically meaningful correspondences, as reflected in its stronger performance. We hope this comparison clearly demonstrates JADE’s advantages over SpotScape.
> > >
> > > Table A: Comparion between JADE and SpotSpace.
> > >
> > > | **Dataset**       | **ARI A (JADE / SpotScape)** | **ARI B (JADE / SpotScape)** | **ACC (JADE / SpotScape)** |
> > > |-------------------|:----------------------:|:----------------------:|:----------------------:|
> > > | **DLPFC**         | **0.551** / 0.407      | **0.550** / 0.434      | **0.780** / 0.773      |
> > > | **Axolotl**       | **0.375** / 0.355      | **0.532** / 0.406      | **0.421** / 0.420      |
> > > | **MERFISH**       | **0.504** / 0.346      | **0.538** / 0.265      | **0.706** / 0.690      |
> > > | **Breast Cancer** | **0.433** / 0.277      | **0.230** / 0.179      | **0.336** / 0.312      |

---

### Official Review · Reviewer_TnAY · 2025-07-02

**Clarity:** 3
**Significance:** 3
**Originality:** 3
**Rating:** 4
**Confidence:** 3

**Summary:**

This paper introduces JADE, a computational framework that simultaneously learns spatial alignments and low-dimensional embeddings for multi-slice spatial transcriptomics data. The method combines graph autoencoders, attention-based OT for alignment, and contrastive learning for embedding refinement. The authors evaluate JADE on human dorsolateral prefrontal cortex (DLPFC) and axolotl brain datasets, claiming superior performance over existing methods in spatial domain detection, alignment accuracy, and batch effect correction.

**Questions:**

1. Have the authors considered extending the framework to handle more than two slices simultaneously using multi-marginal optimal transport?
2. How does the method perform when slices have significantly different numbers of spots or coverage areas?

**Ethical Concerns:**

["NO or VERY MINOR ethics concerns only"]

**Final Justification:**

- The rebuttal clarified key concerns, particularly around the unequal number of spots across slices
- I agree with other reviewers that the novelty of the method is limited

**Limitations:**

Limitations are partially discussed in the Discussion section. More limitations are discussed in the weakness review above

**Quality:**

3

**Strengths And Weaknesses:**

- Strength
  1. The problem of joint optimization of alignment and representation learning is well-motivated. JADE addresses this important problem while other existing methods handle it separately.
  2. The roundtrip loss objective that alternates between alignment and embedding refinement is interesting.
  3. The paper is well written and the experiments are detailed and comprehensive.

- Weakness
  1. JADE assumes full correspondence which might not hold. By enforcing a doubly stochastic alignment matrix with uniform marginals, JADE essentially assumes every location in slice A maps to something in slice B. This is a strong assumption that breaks down if slices aren’t perfectly overlapping or equal sized. Methods like PASTE2[1] handle this by allowing unassigned mass and JADE currently cannot. This is a significant weakness for scalability to larger or diverse datasets
  2. Despite the loss being interesting, JADE’s loss function is complex and combines five functions with their own hyperparameters. The performance is highly sensitive to these hyperparameter choices, and likely requires dataset-specific tuning (Supp section D). I'm concerned about the method's robustness and practical usability

minor typo: line 9, embedding representation of each slice -> embedding representation of each spot

[1] Liu, Xinhao, Ron Zeira, and Benjamin J. Raphael. "PASTE2: partial alignment of multi-slice spatially resolved transcriptomics data." bioRxiv (2023).

---

> ### Author Rebuttal · Authors · 2025-07-31
>
> >  ***Comment:** JADE assumes full correspondence*
>
> **Response:** Thank you for this valuable comment. We acknowledge that JADE's assumption of full alignment, a limitation noted in our Conclusions section. However, full alignment remains a relevant and effective approach in many spatial transcriptomics integration contexts. JADE is designed to exploit this assumption through a principled and scalable framework, and we provide the following clarifications and justifications.
>
> 1. **Full alignment remains appropriate in many practical scenarios**: The assumption of full correspondence is supported by how spatial transcriptomics data are typically generated. In many studies, especially those involving serial or adjacent tissue sections from the same block, standardized protocols and spatial orientation preserve anatomical continuity. This setup, common with platforms such as 10x Visium, Slide-seqV2, HDST, or Stereo-seq, is widely used in applications such as brain mapping, tumor profiling, and developmental tissue analysis [1]. In these contexts, full alignment facilitates accurate reconstruction of tissue structure and is often preferable to under-aligned or sparsity-enforced approaches. Even when integrating data across individuals or datasets, alignment is typically applied to anatomically matched regions (e.g., the same cortical layer or tumor zone), where substantial structural overlap supports biologically meaningful comparison.
> 2. **Full alignment is preferred over partial alignment when overlap is high**: In cases of substantial or complete overlap, full alignment offers advantages over partial matching by avoiding under-alignment. Recent methods such as GPSA [2] and BiGATAE [3], assuming full alignment, demonstrate superior performance in tasks such as multi-slice spatial domain detection and 3D tissue reconstruction. **In contrast**, partial alignment methods, such as PASTE2, are designed to handle substantial missing regions or non-overlapping areas but may underperform or over-regularize when full correspondence is present. Additionally, these methods require tuning a hyperparameter (overlap percentage between the two slices to align), which is challenging and demands significant prior knowledge, often unavailable in practice.
> 3. **JADE can handle imperfect overlaps and unequal size data in practice:** While JADE assumes full alignment, it has demonstrated robustness in real datasets where slices are not perfectly overlapping or equally sized. For example, in the experiments where we evaluated JADE's performance on the DLPFC datasets when the number of spatial locations are unequal across slices (details see **Table R3**), we found that JADE **remains highly stable**. This suggests that JADE is tolerant to moderate mismatch.
> 4. **Fast-JADE supports scalability to large and diverse datasets:** In our paper, we have developed Fast-JADE, a computationally optimized variant. In our benchmarks, Fast-JADE achieves a runtime of 0.015 seconds/epoch for 10,000 spots, scaling linearly to 0.177 seconds/epoch for 100,000 spots, demonstrating strong efficiency for integration at scale.
> 5. **JADE is more than just an alignment tool:** Unlike PASTE or PASTE2, which focus on alignment without embeddings, JADE jointly optimizes alignment and representation in a shared latent space via a roundtrip framework, enhancing both tasks mutually. This enables broader downstream tasks like cell state mapping, spatial domain detection, and trajectory inference.
> 6. **Future directions: relaxing the full correspondence assumption:** We agree that extending to partial alignment would enhance JADE's flexibility. A promising direction is relaxing the doubly stochastic constraint using unbalanced optimal transport, allowing partial matching or unassigned mass. We view this as a valuable extension and will include this discussion in the revised manuscript.
>
> [1] Khan, M., Arslanturk, S., & Draghici, S. (2025). A comprehensive review of spatial transcriptomics data alignment and integration. Nucleic Acids Research, 53(12), gkaf536.
>
> [2] Jones, A., Townes, F. W., Li, D., & Engelhardt, B. E. (2023). Alignment of spatial genomics data using deep Gaussian processes. Nature methods, 20(9), 1379-1387.
>
> [3] Tao, Y., Sun, X., & Wang, F. (2024). BiGATAE: a bipartite graph attention auto-encoder enhancing spatial domain identification from single-slice to multi-slices. Briefings in Bioinformatics, 25(2), bbae045.
>
> >  ***Comment:** Hyperparameters of JADE’s loss. Robustness of JADE*
>
> **Response:**  Thank you for this constructive comment.
>
> To clarify, while there are four hyperparameters ($\lambda_{2}-\lambda_5$) in our loss function, three are fixed across datasets: $\lambda_2$=10 (reconstruction, following GraphST [1]), $\lambda_4$=0.1 (alignment), and $\lambda_5$=1 (marginal), based on empirical validation during development. Only $\lambda_3$ (maintenance) requires tuning (from 0.2 to 2.0), and we employ a **data-driven approach**: if two slices are similar, we use a larger $\lambda_3=2$ to encourage information sharing; otherwise, we use a smaller $\lambda_3=0.2$ to prevent negative transfer. **So the only hyperparameter that needs to be selected is $\lambda_3$.**
>
> Importantly, as shown in Figure 4 (Appendix), JADE’s performance is robust across a broad range of $\lambda_3$ values. Both the average ARI and alignment score show stable results over different $\lambda_3 \in [0.2, 2]$, with only marginal variation. This indicates that the method is not highly sensitive to the precise choice of $\lambda_3$, thus reducing the burden of hyperparameter tuning in practice.
>
> In addition, to further demonstrate the robustness and practical utility of our method, we evaluated JADE on **two additional datasets**: the MERFISH dataset and the Breast Cancer dataset. Please see **Table R5** in our response to Reviewer c7f2 for detailed results. In summary, JADE continues to outperform competing methods on both datasets, **further supporting the robustness and generalizability of our approach**.
>
> [1] Yahui, L., Kok, A., & Jinmiao, C. (2023). Spatially informed clustering, integration, and deconvolution of spatial transcriptomics with graphst. Nature Communications, 14(1):115
>
> >  ***Comment:** minor typo: line 9*
>
> **Response:** Thank you for pointing this out. We will correct this typo in the revised manuscript.
>
> >  ***Comment:** Have the authors considered extending the framework to handle more than two slices simultaneously using multi-marginal optimal transport (MMOT)?*
>
> **Response:**  Thank you for this insightful suggestion. We agree that while we have focused the current version of JADE on the pairwise setting for conceptual clarity and computational efficiency, extending JADE to handle joint alignment of more than two slices is a **natural and promising direction**. We are actively exploring this direction for future extensions of the framework, however, incorporating MMOT into JADE introduces several nontrivial **challenges**.
>
> 1. **Curse of dimensionality:** The coupling tensor $\Gamma \in \mathbb{R}^{n_1 \times n_2 \times \cdots \times n_m}$ grows rapidly with $m$ and the number of points per slice ($n_t$). For example, even with only three slices and $100$ spatial locations per slice, the tensor has $10^6$ entries; for four slices, this rises to $10^8$. This rapid growth poses memory and storage challenges for real datasets with thousands or hundreds of thousands of spatial locations per slice.
> 2. **Expensive computational complexity:** The computational cost of standard MMOT is $\textbf{polynomial in}$ $n^m$ (e.g., $(n^m)^{2}$). Even with efficient variants such as entropic-regularized MMOT using Sinkhorn-based methods, the cost remains approximately linear in $n^m$.
> 3. **Requires careful design of cost function**: In the pairwise setting, cost functions such as Euclidean distance or feature similarity are well-established and straightforward to implement. However, in the multi-slice framework, defining a multi-way cost function that effectively encodes geometric and biological relationships across all slices presents a significant challenge. Conducting pairwise comparisons between every two slices becomes computationally prohibitive as the number of slices increases, necessitating a more efficient approach. One potential strategy is to compute a barycenter (a central representative measure) and map all slices to this barycenter, which could reduce complexity. However, this requires careful design to ensure the barycenter accurately reflects the shared structure and biological context across slices. We leave this for future directions.
>
> We will add these discussions to the revised manuscript.
>
> >  ***Comment:** How does the method perform when slices have significantly different numbers of spots or coverage areas?*
>
> **Response:**  Thank you for this insightful question.  To address this, we conducted additional experiments using the DLPFC dataset (Sample III, slices A and B), where we randomly mask a subset of spots from slice A to simulate scenarios with unequal spot counts. **Table R3** shows the results of varying the proportion of removed spots (e.g., 10%, 25%, 50%) for DLPFC data.
>
> As we can see, across all tested levels of spot removal (up to 50%), both ARI (for Slice A and B), overall alignment accuracy (ACC), and batch correction (iLISI) **remain highly stable**, with only minimal fluctuations. This robustness indicates that JADE effectively handles scenarios with significantly different spot counts between slices, making it practical for real-world applications where tissue sections may have varying sizes.
>
> Table R3: Performance of unequal number of spots on DLPFC dataset.
>
> | % Spots Removed (Slice A)  | ARI (Slice A) | ARI (Slice B) | ACC  | iLISI |
> |:---------------:|:------:|:-------------:|:-----:|:-----:|
> |0%|0.62|0.65|0.83|1.98|
> |10%|0.60|0.66|0.81|1.97|
> |25%|0.61|0.65|0.82|1.98|
> |50%|0.62|0.67|0.82|1.98|

---

> > ### Comment · Reviewer_TnAY · 2025-08-05
> >
> > Thanks for the detailed rebuttal. The additional experiment with unequal spot counts (Table R3) showing stable performance up to 50% removal is pretty convincing. I have updated my review accordingly.

---

> > > ### Author Response · Authors · 2025-08-05
> > >
> > > Thank you for your constructive comments, which helped highlight the strength of our method and improved our manuscript. We deeply appreciate your support.

---

### Official Review · Reviewer_D17F · 2025-07-02

**Clarity:** 3
**Significance:** 3
**Originality:** 3
**Rating:** 5
**Confidence:** 3

**Summary:**

Authors introduce JADE for simultaneous alignment and representation learning of ST spots across multiple paired slices. The objective uses a combination of recon, OT-alignment loss and contrastive loss and is benchmarked against 6 other baselines (either alignment or embedding) on 2 datasets, showing improvement in alignment accuracy and domain segmentation.

**Questions:**

Given the availability of paired H&E image with visium and the long standing literature in image-based registration. How does the authors' approach compared against a computer vision-based baseline of image registration, both in terms of performance and runtime?

**Ethical Concerns:**

["NO or VERY MINOR ethics concerns only"]

**Final Justification:**

I thank the authors for an exceptionally thorough and convincing rebuttal. They have fully addressed my primary concern regarding the lack of comparison to methods that use H&E histology images.

**Limitations:**

Yes.

**Quality:**

4

**Strengths And Weaknesses:**

Strengths
Method is intuitive with good performance. Authors also provide a computationally-efficient approximation.

Weaknesses
Does not take into account H&E images associated with Visium data / lack of comparison with methods that perform image-based alignment.

---

> ### Author Rebuttal · Authors · 2025-07-31
>
> >  ***Comment:** Does not take into account H&E images associated with Visium data / lack of comparison with methods that perform image-based alignment.*
>
> **Response:** We thank the reviewer for this thoughtful comment.
>
> **JADE is designed as a joint alignment and embedding framework**: We would like to respectfully clarify that JADE is not solely an alignment method. Instead, it is designed as a unified framework that simultaneously performs cross-slice spatial alignment and shared low-dimensional representation learning. This joint optimization is central to JADE's design and enables a range of downstream analyses such as spatial domain detection, trajectory inference, and spatially variable gene detection analysis that go beyond alignment-only methods can support. To our knowledge, JADE is the first method to offer this capability within a single framework.
>
> **Rationale for benchmarking choices:** In our original submission, we selected strong and representative alignment methods as baselines. In particular, PASTE was included because of its demonstrated accuracy in alignment tasks, as supported by the recent large-scale benchmarking study by Hu et al. (Genome Biology, 2024). Among the five alignment methods assessed in that study, PASTE was consistently among the top performers.
>
> **New benchmarking comparison with method incorporating H&E images:** We appreciate the reviewer’s suggestion to compare JADE with methods that utilize H&E images. To this end, we have conducted a new benchmarking experiment with GPSA (Jones et al., Nature Methods, 2023), which, to our knowledge, is the only method mentioned in the Hu et al. benchmarking paper that supports the incorporation of both gene expression and histology features for alignment. As shown in Table R2, JADE consistently outperforms GPSA in alignment accuracy across all slice combinations in the DLPFC dataset. These results suggest that JADE is able to achieve strong performance even without directly incorporating image data.
>
> **Advantages and broader applicability of JADE:** While image-based features can in some cases aid alignment, their utility depends heavily on the quality, resolution, and consistency of the imaging data. In practice, H&E images may be noisy, poorly aligned across slices, or contain staining artifacts, and are often collected from adjacent rather than identical tissue sections, limiting their spatial correspondence with the transcriptomic data. Incorporating such features directly into computational models can introduce spurious signals or additional noise, particularly when the image modality does not clearly delineate anatomical structures relevant for molecular alignment. By contrast, JADE is designed to model on gene expression and spatial location information, which are more consistently measured across spatial transcriptomics platforms. This enable JADE to be broadly applicable to a wide range of technologies, such as the Stereo-seq platform (as we shown in Figure 3), where histological imaging is unavailable, and to maintain robustness in settings where image quality is variable or inconsistent.
>
> **Future directions:** Importantly, the JADE framework is modular and can be extended to incorporate image-derived information in future work. For example, histological features could be integrated by modulating the spatial graph, such as assigning image-informed weights to graph edges, or by combining image embeddings with expression-based representations. These extensions could further enhance JADE's utility in settings where high-quality image data are available, while preserving its core advantage of joint alignment and representation learning. Unlike alignment-only methods such as GPSA or PASTE, JADE uniquely supports both spatial alignment and shared low-dimensional embeddings, enabling a broader range of downstream analyses, including clustering, visualization, and trajectory inference.
>
> Table R2: Performance comparison between JADE and GSPA on the DLPFC dataset (Sample I, II, III, slices A, B, C, and D). Bold values indicate the best performance for each metric.
>
> | Sample  | Case | JADE | GSPA |
> |:------:|:----:|:----:|:----:|
> | Sample I | AB   |  **0.76**    |  0.19    |
> |        | BC   |  **0.54**    |  0.20    |
> |        | CD   |  **0.81**    |  0.25    |
> | Sample II | AB   |  **0.88**    |  0.42    |
> |        | BC   |  **0.76**    |  0.34    |
> |        | CD   |  **0.84**    |  0.28    |
> | Sample III | AB   |  **0.83**    |  0.18    |
> |        | BC   |  **0.82**    |  0.17   |
> |        | CD   |  **0.79**    |  0.17    |
> | Average|      |  **0.78**        |   0.24       |
>
> >  ***Comment:** Given the availability of paired H&E image with visium and the long standing literature in image-based registration. How does the authors' approach compared against a computer vision-based baseline of image registration, both in terms of performance and runtime?*
>
> **Response:** We thank the reviewer for this thoughtful comment. We would like to clarify the distinction between our method and conventional image registration approaches, and explain why such baselines are not well suited for the problem that JADE is designed to solve.
>
> **Limitations of histology-based registration in practice:** (1) The feasibility of using H&E image-based registration depends strongly on the spatial transcriptomics workflow. In the original 10x Visium protocol, the H&E image is often obtained from an adjacent tissue section, not the same one used for RNA capture. This is because H&E staining can compromise RNA quality, and separate serial sections are used to preserve transcriptomic integrity. As a result, aligning the transcriptomic data with the corresponding H&E image already requires an additional registration step, and introduces unavoidable discrepancies between the molecular and histological data. (2) Recently, the Visium CytAssist workflow enables H&E imaging and transcript capture from the same tissue section. This allows for more precise correspondence between image and gene expression data at the spot level. However, even under this setup, a standard image registration pipeline has several important limitations when applied to the task of multi-sample spatial transcriptomics integration. As mentioned in our previous response, even when histological information is incorporated into the alignment process, as is the case with GPSA, the resulting performance is substantially worse than that of JADE. This might suggest that approaches relying exclusively on histological images are even less likely to succeed. Moreover, many spatial transcriptomics platforms, including the Stereo-seq technology used in Figure 3 of our manuscript, do not provide histological images. These considerations highlight JADE’s broader applicability and robustness across different spatial transcriptomics technologies.
>
> **Image registration is insufficient for molecular alignment:** (1) Image registration methods rely exclusively on histological features and ignore gene expression profiles entirely. They are therefore sensitive to variations in staining intensity, tissue processing, and imaging conditions, which can differ across experiments and reduce registration accuracy. (2) Image-based alignment cannot be applied to cross-platform or cross-modality settings, where histology images may not exist or may not be comparable. (3) Image registration does not address the core challenge of transcriptomic alignment, which involves learning a biologically meaningful correspondence between spatially resolved molecular profiles across samples.
>
> **JADE provides embeddings for integrated molecular analysis:** Most importantly, image registration methods do not generate any molecular embedding for the spatial transcriptomics data. They simply return a spatial transformation that aligns two tissue sections at the image level, without learning a latent representation of gene expression. In contrast, JADE jointly aligns multiple tissue sections and learns a low-dimensional embedding that reflects shared transcriptional and spatial structure across samples. These embeddings are directly useful for downstream analyses such as clustering, visualization, and trajectory inference, and provide a richer representation than spatial alignment alone.
>
> We hope this clarifies the distinct goals and advantages of JADE relative to conventional image registration.

---

### Official Review · Reviewer_Yzhq · 2025-07-04

**Clarity:** 3
**Significance:** 3
**Originality:** 3
**Rating:** 4
**Confidence:** 4

**Summary:**

In this paper, the authors proposed  a unified computational framework to simultaneously learns spatial alignments and their embeddings across pairs of tissue slices. The spatial information of each slide was initially represented as k-nearest neighbor graph and embedding was generated via graph convolutional networks (GCNs). Graph attention was then employed to obtain alignment using the embeddings. The final objective function includes self-supervised graph contrastive loss to maintain local neighborhood structure, alignment loss, gene expression reconstruction  loss and a regularization term.

The authors performed extensive experiments using two datasets and compared their performance with existing approaches in terms of alignment accuracy and downstream applications. Results demonstrated its superior performance.

**Questions:**

1. The approach can only work  on two slices. Can the authors do some experiments, for example, first establish alignments between slices A&B, and B&C, derive alignment between A&C, and compare it with the direct alignment of A&C?

**Ethical Concerns:**

["NO or VERY MINOR ethics concerns only"]

**Limitations:**

Yes.

**Quality:**

3

**Strengths And Weaknesses:**

Strengths
1. The paper addresses a timely problem in bioinformatics. It is well written and easy to follow.
2. The idea of performing alignment and learning the embedding simultaneously is novel.
3. The results are convincing.

Weaknesses
1. The proposed approach does not differentiate spatial from temporal domain, as showed in the two different datasets, which could be very different in terms of the actual corresponding locations between the two slices.
2. Using graphs to represent the locations will lose coordinate information. Can the authors access the impact of such a representation?
3. It can only work for a pair of slices.

---

> ### Author Rebuttal · Authors · 2025-07-31
>
> We thank the reviewer for the thoughtful and constructive comments. Please see below for our response.
>
> >  ***Comment:** The proposed approach does not differentiate spatial from temporal domain, as showed in the two different datasets, which could be very different in terms of the actual corresponding locations between the two slices.*
>
> **Response:** Thank you for raising this important point. JADE is designed to align spatial transcriptomics (SRT)  datasets across tissue sections without assuming any temporal ordering or anatomical continuity. This design reflects the current structure of most publicly available spatial omics datasets.
>
>
> **Spatiotemporal SRT data are still rare**: While modeling temporal structure can be important for studying dynamic biological processes, spatiotemporal spatial transcriptomics datasets, which capture spatially resolved gene expression over time, are currently limited due to technical challenges. These include difficulties in collecting consistently aligned serial sections across time points, preserving spatial structure, and maintaining high RNA quality throughout the time course.
>
> **Focus on multi-sample integration:** In contrast, multi-sample spatial transcriptomics datasets are much more widely available. These include data collected across individuals, biological replicates, or conditions, typically without consistent spatial or temporal alignment. JADE is specifically designed for this general setting. It learns a shared embedding by leveraging both spatial context and gene expression similarity, without relying on external spatial registration or temporal metadata. This enables robust alignment across diverse datasets such as tissue atlases, cross-sectional disease studies, and multi-region profiling.
>
> **Effectiveness in cross-stage and heterogeneous data:** Importantly, even in datasets spanning different developmental stages, such as the Axolotl Telencephalon Dataset from Stereo-seq (as shown in Figure 3 in our original submission) are drawn from different individuals and contain distinct spatial resolutions and morphologies. Nevertheless, JADE is able to successfully integrate these samples by leveraging both spatial context and gene expression similarity. In our experiments, JADE achieved the highest domain detection accuracy and alignment performance across juvenile and adult axolotl brain slices. It was the only method to reconstruct key anatomical structures (e.g., the peripheral vascular leptomeningeal cell ring and Nptx+ lateral pallium excitatory neuron), yielded the most coherent latent embeddings with strong mixing across slices, and identified biologically meaningful spatial expression patterns for marker genes such as *SCGN*, *ZIC1*, and PENK. These results highlight that even without explicitly modeling temporal structure, JADE can align cross-stage datasets by leveraging gene expression signals that implicitly reflect developmental progression.
>
> **Future extension to temporally resolved data:** While modeling true temporal progression remains an important goal for understanding dynamic biological processes, current spatial transcriptomics technologies do not support dense time-course sampling from the same tissue. As temporally resolved spatial datasets become more accessible, the JADE framework could be extended to model temporal progression by integrating trajectory-informed regularization, time-aware embeddings, or spatiotemporal graph structures. These extensions would enable JADE to explicitly capture both spatial and temporal dependencies in future datasets. We appreciate the reviewer’s suggestion and we will include these discussions in the revised manuscript.
>
> >  ***Comment:** Using graphs to represent the locations will lose coordinate information. Can the authors access the impact of such a representation?*
>
> **Response:** Thank you for your valuable comment.
>
> **Use of spatial graphs in prior work**: Constructing spatial graphs from location coordinates is a standard and well-validated approach in methods for modeling spatial transcriptomics data (e.g., GraphST [1], STAGATE [2], SpaGCN [3]). These methods prioritize capturing local neighborhood relationships, such as spatial proximity and context, over absolute coordinate values, which can be distorted across sections or individuals due to variations in tissue placement or morphology. Thus, spatial graphs provide a robust mechanism to borrow strength from nearby locations while being resilient to global spatial inconsistencies.
>
> **Empirical evaluation**: To directly address the reviewer’s concern, we conducted an additional experiment in which we concatenated the original spatial coordinates ($S$) to the gene expression matrix ($X$), resulting in an augmented feature matrix $X' \in \mathbb{R}^{n \times (p+2)}$ as the new input for each spatial location.  We set the adjacency matrix $A$ to the identity matrix, thereby removing all graph-based connectivity and relying solely on the absolute spatial positions. We then ran JADE on this modified input (We refer to this procedure as **JADE-concat**). As shown in **Table R1**, this brute-force inclusion of absolute coordinates performs worse than our original JADE framework, which leverages the spatial graph, across all evaluation metrics, including Adjusted Rand Index (ARI), Alignment Accuracy (ACC), and integrated local inverse Simpson’s index (iLISI). This shows the effectiveness of using spatial graph.
>
> Table R1: Performance comparison between JADE and JADE-concat on the DLPFC dataset (Sample III, slices A and B). Bold values indicate the best performance for each metric.
>
> |                 | **ARI (Slice A)** | **ARI (Slice B)** | **ACC** | **iLISI** |
> | --------------- | :-----------------: | :-----------------: | :--------------------: | :--------------------------: |
> | JADE        |     **0.62**                |     **0.65**                |                 **0.83**       |    **1.98**                          |
> | JADE-concat |     0.58                |       0.51              |               0.66         |  1.91                           |
>
>
> >  ***Comment:** It can only work for a pair of slices. The approach can only work on two slices. Can the authors do some experiments, for example, first establish alignments between slices A&B, and B&C, derive alignment between A&C, and compare it with the direct alignment of A&C?*
>
> **Response:** Thank you for this insightful comment. Following your suggestions, we conducted additional experiments on the DLPFC dataset (Sample III) using slices A, B, and C. Specifically, we first computed pairwise alignments $\Pi_{AB}$ (between A and B) and $\Pi_{BC}$ (between B and C) using JADE, then derived a transitive alignment $\Pi_{AC} \propto \Pi_{AB} \times \Pi_{BC}$ (followed by Sinkhorn normalization to enforce doubly-stochastic properties). We compared this transitive $\Pi_{AC}$ against the direct alignment obtained by running JADE on A and C. We found that the alignment accuracy for the direct A–C alignment was **0.767**, whereas the transitive alignment via A–B–C yielded an accuracy of **0.749**. This result suggests that although transitive alignment using JADE is feasible, it tends to accumulate intermediate alignment noise, leading to slightly lower accuracy than direct pairwise alignment. We appreciate the reviewer’s suggestion, which highlights an important point: JADE's pairwise design supports flexible alignment across arbitrary slice pairs and enables indirect mapping when needed. However, direct alignment remains the preferred strategy due to its superior accuracy, as demonstrated in our experiments. We will include this discussion and analysis in the revised manuscript.
>
> [1] Yahui Long et al. GraphST: Spatially informed clustering, integration, and deconvolution of spatial transcriptomics. Nat. Commun. 14, 1155 (2023).
>
> [2] Kangning Dong and Shihua Zhang. Adaptive graph attention auto-encoder for spatial domain detection in transcriptomics. Nat. Commun. 13, 1739 (2022).
>
> [3] Jian Hu et al. SpaGCN: Integrating gene expression, spatial location, and histology via graph convolutional networks. Nat. Methods 18, 1342–1351 (2021).

---

### Decision · Program_Chairs · 2025-09-17

**Decision:**

Accept (poster)

**Comment:**

This paper introduces JADE, a framework that jointly learns spatial alignments and shared embeddings across tissue slices using graph neural networks, attention-based OT, and contrastive learning. Evaluations on DLPFC and Axolotl datasets, with additional rebuttal experiments, show consistent improvements over existing methods.

### Strengths
- Addresses an important and timely problem in spatial transcriptomics.
- Joint optimization of alignment and embedding is a well-motivated and practically useful idea.
- Clear writing and thorough experiments, including new analyses in rebuttal (heterogeneous datasets, runtime scaling, comparisons with image-based and graph-based baselines).
- Demonstrates strong and robust performance across multiple datasets and tasks.

### Weaknesses
- Methodological novelty is incremental, drawing on existing ideas from GraphST and PASTE.
- Assumption of full correspondence may limit applicability in some scenarios.
- Evaluation breadth, while expanded in rebuttal, could still be more comprehensive.

Despite some limitations in novelty and scope, the paper makes a solid and practically impactful contribution, with convincing results and thoughtful rebuttal clarifications. I recommend acceptance.